# DyLLM: Efficient Diffusion LLM Inference via Saliency-based Token Selection and Partial Attention

**Younjoo Lee** [1]  **Seungkyun Dan** [1]  **Junghoo Lee** [1]  **Jaiyoung Park** [1]  **Jung Ho Ahn** [1]

## Abstract

Masked diffusion language models enable parallel token decoding, providing a promising alternative to the sequential nature of autoregressive generation. However, their iterative denoising process remains computationally expensive because it repeatedly processes the entire sequence at every step. We observe that across these diffusion steps, most token representations remain stable; only a small subset, which we term *salient tokens*, contributes meaningfully to the next update. Leveraging this temporal sparsity, we present **DyLLM**, a training-free inference framework that accelerates decoding by selectively computing only these salient tokens. DyLLM identifies saliency by measuring the cosine similarity of attention contexts between adjacent denoising steps. It recomputes feed-forward and attention operations only for salient tokens while reusing cached activations for the remainder. Across diverse reasoning and code-generation benchmarks, DyLLM achieves up to 9.6× higher throughput while largely preserving the baseline accuracy of representative open-source diffusion LLMs, LLaDA and Dream.

## 1. Introduction

The emergence of discrete denoising diffusion probabilistic models (D3PM) (Austin et al., 2021a) has introduced a new paradigm for language modeling by enabling diffusion processes directly within discrete token spaces via masking. This paradigm shift has led to the development of high-performance *Masked Diffusion Language Models* (MDLMs) such as LLaDA (Nie et al., 2025), Dream (Ye et al., 2025), and Gemini Diffusion (Google DeepMind, 2025), which are now approaching the performance lev-

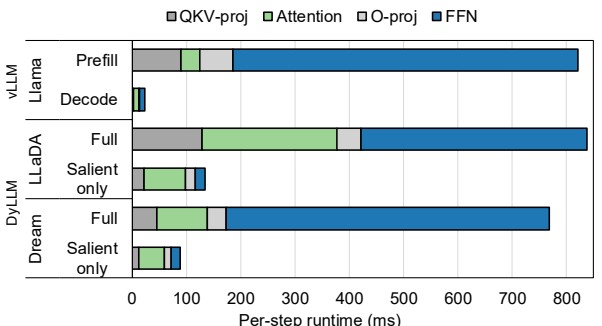

*Figure 1.* Runtime breakdown of autoregressive decoding with vLLM (Kwon et al., 2023), original diffusion LLM implementation, and DyLLM on random GSM8K 5-shot prompts (batch size = 16). Original diffusion LLM repeats the full steps, whereas DyLLM recomputes only salient tokens, reducing the dominant per-step overhead.

els of standard *Autoregressive Language Models* (ARLMs), such as Llama (Grattafiori et al., 2024).

Unlike ARLMs, which are constrained by a sequential, token-by-token generation process, MDLMs lift this limitation by initializing the response as a sequence of mask tokens and leveraging bidirectional attention to predict the entire sequence simultaneously. By iteratively unmasking tokens based on the model's predicted probabilities, an MDLM decodes multiple tokens in parallel, providing a viable path toward superior generation throughput.

Despite these advantages, the iterative refinement of underlying diffusion-based decoding introduces a significant computational bottleneck. In ARLMs, execution follows a strict sequential order, which naturally enables incremental Key-Value (KV) caching (Pope et al., 2023); only the newly generated token requires computation at each step. In contrast, MDLMs operate without such a fixed spatial ordering. Because bidirectional attention updates the global context during each refinement step, the model must repeatedly process the full sequence.

Consequently, denoising steps in an MDLM resemble a "repeated prefill" operation, leading to prohibitive computational waste as the number of iterations increases. Figure 1 highlights a runtime perspective that clarifies this trade-off. While ARLMs exhibit highly efficient decoding steps after

[1]Seoul National University, Seoul, Republic of Korea. Correspondence to: Younjoo Lee <younjoo0614@snu.ac.kr>, Jung Ho Ahn <gajh@snu.ac.kr>.

*Proceedings of the 43rd International Conference on Machine Learning*, Seoul, South Korea. PMLR 306, 2026. Copyright 2026 by the author(s).

the initial prefill, MDLMs incur a heavy compute tax at every step, primarily dominated by Feed-Forward Network (FFN) operations.

Prior works, including Fast-dLLM (Wu et al., 2026a), dKV-Cache (Ma et al., 2025), SparseD (Wang et al., 2026), dLLM-Cache (Liu et al., 2025), and Elastic-Cache (Nguyen-Tri et al., 2026), have explored the acceleration of MDLM inference. Among these, caching-based techniques have emerged as a prominent area of investigation, aiming to reduce redundant computations across steps. However, Fast-dLLM and dKV-Cache rely on fixed caching strategies that exploit the sequential structure of language but fail to capture the diffusion-LLM-specific dynamics of layer-wise variation in representation stability. Similarly, dLLM-Cache and Elastic-Cache utilize activation similarity to trigger cache updates, but these methods do not fully exploit the layer-wise and token-wise heterogeneity of representation stability during inference steps.

In this paper, we propose **DyLLM**, a training-free inference framework that accelerates MDLM decoding by exploiting layer-wise temporal sparsity. Our work is built on the observation that across consecutive diffusion steps, the majority of token representations remain stable (Ma et al., 2025; Wu et al., 2026a; Liu et al., 2025); only a small subset, which we define as *salient tokens*, undergoes meaningful transitions that contribute to the next update.

DyLLM identifies these salient tokens by measuring the cosine similarity of the attention contexts between adjacent steps. Leveraging this property, DyLLM recomputes FFN and attention operations only for salient tokens while reusing cached results for the remaining tokens. To further mitigate the quadratic overhead of attention, we introduce a saliency-aware approximate sparse attention scheme that approximates context updates for stable tokens with high fidelity. Figure 1 shows that DyLLM substantially alleviates the dominant bottleneck, leading to improved throughput in salient-only steps. Code is available at https://github.com/scale-snu/DyLLM.git.

Our key contributions are three-fold:

- Layer-adaptive saliency selection: We introduce a dynamic token selection policy that identifies salient tokens at each layer, allowing the model to bypass redundant FFN computations for stable hidden states.

- Saliency-aware approximate attention: We propose an approximate attention scheme that exploits activation sparsity to eliminate redundant context updates, reducing the complexity of the attention operation.

- Scalable sparse inference: We demonstrate that DyLLM scales robustly with increasing parallel decoding degrees ($n_u$). Across LLaDA and Dream models,

DyLLM achieves up to $7.6\times$ and $9.6\times$ higher throughput while preserving baseline accuracy across reasoning and code-generation benchmarks.

## 2. Language Model Paradigms

We consider the generation task of language models, where an input prompt $\mathbf{x}^P = [x^1, ..., x^{L_P}]$ of length $L_P$ is given to generate a response $\mathbf{x}^R = [x^{L_P+1}, ..., x^{L_P+L_R}]$ of length $L_R$. Each token $x^l$ belongs to a discrete vocabulary $\mathcal{V} = \{1, ..., V\}$. Following the convention, we use superscript to denote the spatial position $l$ and subscript to denote the diffusion timestep $t \in \{T, ..., 0\}$; e.g., the state of a token at position $l$ and diffusion step $t$ is denoted as $x_t^l$.

To maintain focus on the computational efficiency of the generation process, the following preliminaries focus on the inference phase. We refer the reader to Appendix A for further background on the theoretical foundations of Diffusion LLMs.

### 2.1. Autoregressive Sampling: Sequential and One-Pass

The AR generation process operates in a causal manner, predicting one token at each positional step from left to right. Concretely, given an input sequence, an AR model outputs a categorical distribution over $\mathcal{V}$, conditioned on the previously fixed tokens $\mathbf{x}^{1:l-1}$.

$$x^l \sim p(x|\mathbf{x}^{1:l-1})$$

Once a token $x^l$ is sampled, it is appended to the context and fixed for the remainder of the generation. This mechanism ensures that each token is computed as an output in exactly one forward pass of the model, allowing for highly efficient KV caching. However, the AR process imposes a strict sequential dependency, where the $l$-th forward pass can proceed only after the $l - 1$-th token has been generated. While recent speculative decoding techniques (Leviathan et al., 2023) attempt to mitigate this sequential bottleneck by generating multiple tokens per step, they still operate within the AR paradigm's causal constraints. We consider these approaches beyond the scope of this paper and instead focus on the following diffusion-based paradigm, which enables parallel decoding through iterative refinement.

### 2.2. Diffusion-based Sampling: Parallel and Multi-Pass

In contrast to ARLMs, MDLMs treat generation as a global denoising task, breaking the one-token-per-forward-pass constraint through iterative refinement. The generation process is formulated as a *filling-in-the-blanks* task, where the model starts with a fully masked response and progressively reveals tokens from the masked sequence. The parallel refinement process can be formally described as follows:
**Setup**: Given an input prompt $\mathbf{x}^P$ and a target response $\mathbf{x}^R$, the generation proceeds through a sequence of target states $\mathbf{x}_t^R$ for $t \in \{T, ..., 0\}$. At the initial timestep $T$, the response is initialized as a sequence of mask tokens $[M] \in \mathcal{V}$. Consequently, the initial input to the model is

constructed as:

$$\mathbf{x}_T = [\mathbf{x}^P; \mathbf{x}_T^R] = [x^1, \ldots, x^{L_P}, \underbrace{[\mathbf{M}], \ldots, [\mathbf{M}]}_{L_R \text{ tokens}}]$$

Throughout the sampling process, the prompt $\mathbf{x}^P$ remains invariant, providing a constant context, while the response state $\mathbf{x}_t^R$ is updated over $T$ iterations.

**Refinement and Unmasking**: At each denoising step $t$, the model processes the concatenated sequence $\mathbf{x}_t = [\mathbf{x}^P; \mathbf{x}_t^R]$ to estimate, for every position $r$, the categorical distribution

$$p(\hat{x}^r \mid \mathbf{x}_t), \quad r \in \{1, \ldots, L_P + L_R\},$$

where $\hat{x}^r$ denotes the decoded (unmasked) token predicted at position $r$. In principle, all masked positions can be unmasked in a single denoising step.

Based on these predictions, the response state transitions from $\mathbf{x}_t^R$ to $\mathbf{x}_{t-1}^R$ by replacing a subset of mask tokens with tokens sampled from the predicted probability distributions. The number of tokens revealed at each time step is controlled by a transition schedule $\beta_t \in [0, 1]$, which determines the fraction of positions that remain masked at step $t$.

Given this budget of tokens to be revealed, the specific positions to unmask are selected according to a confidence-based sampling strategy (Nie et al., 2025; Ye et al., 2025). In particular, the model identifies a subset of positions with the highest prediction confidence (e.g., the highest logit values) to be unmasked, whereas the remaining positions are kept as [M] for further refinement in the next step.

Ultimately, the diffusion approach is designed to generate a full response in a constant number of steps. By decoding multiple tokens per forward pass, MDLMs have the potential to significantly reduce the number of forward passes inherently required by the autoregressive paradigm, enabling much higher throughput.

### 2.3. Semi-Autoregressive Decoding: Locally Parallel, Globally Sequential

While MDLMs enable parallel refinement, simultaneously refining the entire sequence makes the denoising process more challenging, often leading to degraded generation quality. Block-wise semi-autoregressive (semi-AR) decoding (Han et al., 2023) addresses this issue by scheduling the decoding process in a left-to-right, block-wise manner. Specifically, the target response is partitioned into fixed-length blocks of size $B$, which are decoded sequentially, while tokens within each block are refined in parallel using the MDLMs' iterative unmasking procedure.

This block-wise decoding schedule progressively fixes previously decoded blocks. Empirically, this strategy yields substantial accuracy improvements (Jazbec et al., 2026); for example, LLaDA's performance on the GSM8K benchmark increases from 58.07 to 77.79 under semi-AR decoding (see Section D.3). This indicates that semi-AR results in better

generation quality by enabling stable prefix representations across refinement steps. As discussed next, this execution structure additionally enables efficient reuse of attention key-value representations through block-wise KV caching.

### 2.4. The Efficiency Dilemma: Decoding Parallelism vs. Caching

Interestingly, the parallel refinement capability of MDLMs introduces a fundamental efficiency dilemma. In AR decoding, KV caching (Pope et al., 2023) is highly effective because previously computed KV representations remain unchanged, and each step only computes a single new position. In contrast, MDLMs recompute every position in every iteration, as the bidirectional attention mechanism requires the entire sequence to be re-processed at every denoising step $t$. This leads to a prohibitive per-iteration computational cost that often offsets the benefits of parallel decoding.

To mitigate this issue, several caching strategies have been proposed, ranging from temporal approaches such as periodic refresh to spatial and fine-grained activation reuse. As an early effort, dKV-Cache (Ma et al., 2025) introduces a periodic refresh strategy that reuses cached KV states across multiple timesteps and refreshes them to correct accumulated errors. More recently, Fast-dLLM (Wu et al., 2026a) restricts the spatial scope of re-computation through block-wise caching. Specifically, its PrefixCache reuses the context of the prompt and previously decoded blocks during refinement, based on the observation that prefix activations remain stable across diffusion steps. Its DualCache further extends this mechanism by additionally reusing the masked suffix tokens and updating them via periodic refresh. This allows the model to compute only the active target block in most iterations, thereby improving throughput with minimal accuracy loss.

In parallel to the block-based methods, another line of work explores token-level adaptive caching. dLLM-Cache (Liu et al., 2025) extends caching to attention outputs and FFN activations, selectively recomputing tokens whose value vectors change the most between steps. Elastic-Cache (Nguyen-Tri et al., 2026) further increases caching flexibility by adaptively determining, based on attention weights, which tokens should participate in the computation.

Despite these advancements, a critical gap remains. Prior works address either spatial, block-wise redundancy or token-wise redundancy, but they often rely on fixed schedules or global thresholds that do not account for layer-wise diffusion dynamics. DyLLM addresses this gap by selecting subsets of tokens adaptively at each layer and denoising step.

## 3. DyLLM

To mitigate the computational redundancy in MDLMs, we propose DyLLM, a framework that adaptively recomputes only tokens whose internal representations change signif-

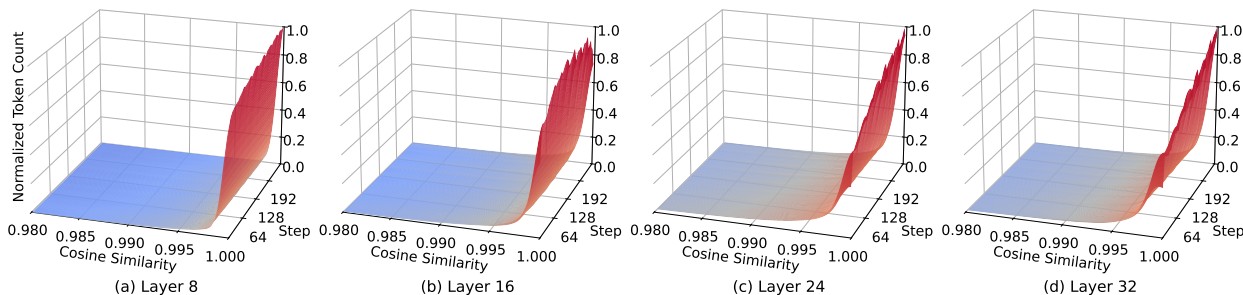

*Figure 2.* Distribution of temporal cosine similarity $s_{t,l}$ across denoising steps and layers ($l \in \{8, 16, 24, 32\}$), measured on LLaDA with GSM8K 5-shot prompts. Most token representations remain highly similar across consecutive steps, with the density concentrated near $s_{t,l} \approx 1$. In deeper layers, the distribution shows a larger low-similarity tail, indicating that more tokens undergo meaningful updates.

icantly between adjacent denoising steps. As illustrated in Figure 1, DyLLM accelerates inference by transitioning from an initial cache warmup phase (Full) to an accelerated refinement phase (Salient only), mirroring the prefill-and-decoding phases in ARLMs. DyLLM is built upon two key insights: (1) *layer-wise temporal sparsity* of MDLM and (2) *position-wise delta propagation* in Transformer blocks.

### 3.1. Characterizing Temporal Sparsity

We first analyze the temporal dynamics of internal representations during denoising. Let $C_{t,l}^{(i)}$ denote the attention context vector for token $i$ at layer $l$ and timestep $t$, computed via softmax-normalized attention scores and value vectors. To quantify the evolution of $C_{t,l}$, we define *temporal cosine similarity* as:

$$s_{t,l}^{(i)} = \frac{C_{t,l}^{(i)} \cdot C_{t-1,l}^{(i)}}{\|C_{t,l}^{(i)}\| \|C_{t-1,l}^{(i)}\|}$$

Figure 2 illustrates the distribution of $s_{t,l}$ across $T = 256$ diffusion steps. Our analysis reveals two critical findings: First, for the majority of tokens, the distribution is heavily skewed toward unity ($s_{t,l}^{(i)} \approx 1.0$) across all layers, indicating that attention contexts remain highly stable between consecutive iterations. Second, the degree of stability is layer-dependent: early layers are dominated by tokens with near-perfect similarity, whereas deep layers show a broader distribution.

### 3.2. Salient Token Selection: Identifying Semantic Deltas

Based on these observations, we define *salient tokens* at step $t$ and layer $l$ as a set of indices $\mathcal{A}_{t,l} = \{i \mid s_{t,l}^{(i)} < \tau\}$, where $\tau$ is a selection threshold. We denote its cardinality by $L_{sal}^{t,l} = |A_{t,l}|$, or simply $L_{sal}$ when the step and layer are clear from context. During timestep $t$, for non-salient tokens ($i \notin \mathcal{A}_{t,l}$), we skip the subsequent FFN computation and reuse the previous state from the FFN output cache, while only salient tokens undergo full re-computation. Furthermore, Propositions 3.1 and 3.2 establish the error bounds of our selection metric. Complete proofs for all propositions

are provided in Appendix E.

**Proposition 3.1** (Scale Invariance under Linear Projection). *Let $W_o \in \mathbb{R}^{d \times d}$ be the output projection matrix, and $\alpha \in \mathbb{R}^+$ be a positive scaling factor. The composite operation of linear projection followed by RMSNorm satisfies:*

$$RMSNorm((\alpha C)W_o) = RMSNorm(CW_o) \quad (1)$$

Proposition 3.1 implies that the magnitude of the attention context $C_{t,l}$ does not affect the input to the subsequent FFN layer; only the directional alignment of the projected vector $C_{t,l}W_o$ matters. Next, we establish the relationship between our metric $s_{t,l}$ and the approximation error in the final normalized output.

**Proposition 3.2** (Error Bound via Directional Alignment). *Let $\delta = \|\text{RMSNorm}(C_{t,l}W_o) - \text{RMSNorm}(C_{t-1,l}W_o)\|_2$ denote the approximation error at layer $l$. Assuming that the output projection $W_o$ is well-conditioned, the error admits the following upper bound:*

$$\delta \leq \kappa(W_o)\sqrt{2(1 - s_{t,l})}, \quad (2)$$

*where $s_{t,l}$ is the temporal cosine similarity and $\kappa(W_o) = \sigma_{\max}(W_o)/\sigma_{\min}(W_o)$ denotes the condition number of $W_o$.*

**Implication.** Proposition 3.2 formalizes that the temporal shift of the FFN input is directly related to our proxy $s_{t,l}$. When a token's representation is near-stationary $s_{t,l}^{(i)} \to 1$, skipping the FFN computation introduces almost no error. By contrast, tokens with low $s_{t,l}^{(i)}$ exhibit a large temporal shift, making them salient for the final generation quality. By thresholding on $s_{t,l}$, DyLLM implicitly controls the error propagation budget across model layers. In early layers, where most positions exhibit low sensitivity, we prune computation aggressively. In deeper, more sensitive layers, our threshold-based policy automatically expands the set of salient tokens to safeguard the final generation quality.

After $T_{full}$ full steps (set to 4 in this work) to initialize and stabilize caches, subsequent steps operate only on salient tokens.

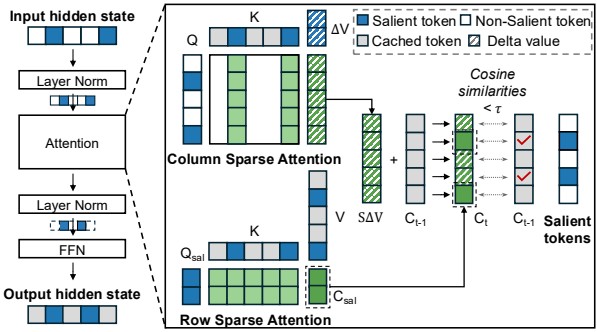

*Figure 3.* Approximate attention in DyLLM. DyLLM computes exact attention rows for salient query tokens using the cached key/value states (row-sparse path). For non-salient query tokens, it approximates context updates using only the attention-score columns corresponding to salient token indices (column-sparse path).

### 3.3. Propagation of Semantic Deltas Across Layers

Beyond bypassing FFN layers, identifying salient tokens enables further optimization of the attention mechanism by avoiding full context recomputation. Since semantic deltas are localized to a sparse subset $\mathcal{A}_{t,l}$, the updates to the attention context can be decomposed and propagated with high efficiency.

Let $\Delta C_{t,l} = C_{t,l} - C_{t-1,l}$, and $\Delta V_{t,l} = V_{t,l} - V_{t-1,l}$ represent the temporal change in the attention context and value matrix at layer $l$ and denoising step $t$. Using the attention score matrix $S_{t,l}$, this delta can be expanded as:

$$
\begin{aligned}
\Delta C_{t,l} &= (S_{t-1,l} + \Delta S)(V_{t-1,l} + \Delta V_{t,l}) - S_{t-1,l}V_{t-1,l} \\
&= (S_{t-1,l} + \Delta S)\Delta V_{t,l} + (\Delta S)V_{t-1,l} \\
&= S_{t,l}\Delta V_{t,l} + (\Delta S)V_{t-1,l}
\end{aligned}
\tag{3}
$$

The decomposition in Equation (3) reveals that temporal updates in one step are propagated to the next layer through two channels: (1) updates to token content via ($\Delta V$), and (2) re-routing of attention weights ($\Delta S$). Based on the active set $\mathcal{A}_{t-1,l}$ from the previous iteration, we execute a dual-path update strategy (Figure 3):

- **Salient Path (Exact Row Update).** For tokens $i \in \mathcal{A}_{t,l-1}$, the representation undergoes a significant transition. We therefore fully recompute the $i$-th row of the attention output matrix $C_{t,l}$, allowing salient tokens to dynamically update their attention patterns. As illustrated by the row-sparse path in Figure 3, this computation is sparse along the query dimension: only the rows corresponding to salient query tokens are recomputed, while each selected query attends to all key/value vectors.

- **Non-salient Path (Approximate Update).** For tokens $i \notin \mathcal{A}_{t,l-1}$, the query remains approximately stationary, implying $\Delta S_{t,l}^{(i,\cdot)} \approx \mathbf{0}$. Accordingly, the update

*Table 1.* Impact of Saliency-based FFN. The "Salient" column reports accuracy when FFN computation is restricted to salient tokens while using full attention. The "Salient+Approx." column reports the full DyLLM configuration, where saliency-aware approximate attention is additionally applied.

| Bench. | Model | Thres. | Orig. | Salient | Salient + Approx. |
|---|---|---|---|---|---|
| GSM8K | LLaDA | 0.995 | 77.79 | 78.09 | 78.01 |
| | | 0.99 | | 78.85 | **79.08** |
| | | 0.985 | | 78.62 | 79.00 |
| | Dream | 0.9975 | 75.59 | **79.98** | 79.30 |
| | | 0.995 | | **79.98** | 78.09 |
| | | 0.99 | | 78.70 | 76.04 |
| MBPP | LLaDA | 0.995 | 29.2 | 26.00 | 28.20 |
| | | 0.99 | | 28.00 | **30.00** |
| | | 0.985 | | 27.60 | 27.60 |
| | Dream | 0.9975 | 54.60 | 56.00 | **56.80** |
| | | 0.995 | | 56.00 | 55.40 |
| | | 0.99 | | 55.20 | 53.20 |
| MATH | LLaDA | 0.995 | 33.22 | 38.28 | **38.68** |
| | | 0.99 | | 38.04 | 38.08 |
| | | 0.985 | | 38.40 | 38.50 |
| | Dream | 0.9975 | 37.60 | **45.54** | 45.12 |
| | | 0.995 | | **45.54** | 43.80 |
| | | 0.99 | | 44.60 | 42.94 |
| MMLU-Pro | LLaDA | 0.995 | 37.31 | **37.57** | 37.06 |
| | | 0.99 | | 37.11 | 36.42 |
| | | 0.985 | | 36.58 | 36.58 |
| | Dream | 0.9975 | 47.94 | **48.97** | 47.45 |
| | | 0.995 | | **48.97** | 47.21 |
| | | 0.99 | | 47.95 | 46.88 |

simplifies to $\Delta C_{t,l}^{(i)} \approx S_{t,l}^{(i,\cdot)}\Delta V_{t,l}$, where the attention weights are effectively reused from the previous iteration. Since $\Delta V_{t,l}$ is row-sparse, i.e., non-zero only for value vectors whose indices belong to $\mathcal{A}_{t,l-1}$, the update only requires the attention score columns associated with salient tokens. Thus, as illustrated by the column-sparse path in Figure 3, non-salient context tokens are updated by gathering contributions only from salient tokens, rather than computing full attention updates induced by both salient and non-salient tokens.

**Computational Efficiency.** This strategy reduces the complexity of attention from $O(N^2d)$ to $O(N \cdot |\mathcal{A}_{t,l-1}|d)$, where $N$ denotes the sequence length and $d$ the attention head dimension. For the stable path, we only fetch columns of $S_{t,l}$ indexed by $\mathcal{A}_{t,l-1}$, avoiding the quadratic cost of a full attention context computation. Since $|\mathcal{A}_{t,l-1}|$ is typically a small fraction of $N$ (Figure 2), this yields substantial throughput gains while maintaining high fidelity (Figure 4).

### 3.4. Impact of Saliency-Aware Inference on Accuracy

Besides computational efficiency, DyLLM preserves model accuracy and yields modest accuracy improvements in several cases. In Table 1, restricting FFN computation to salient tokens consistently preserves and in several cases modestly improves model accuracy across representative benchmarks. These results suggest that selectively recomputing salient

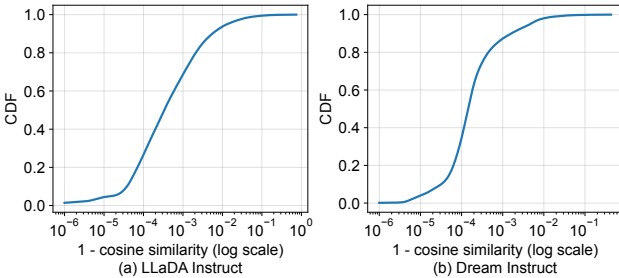

*Figure 4.* Error of approximate attention compared to the exact attention measured by cosine similarity on LLaDA and Dream.

tokens is sufficient to maintain generation quality while substantially reducing redundant computation. A contributing factor is the behavior of softmax normalization. By construction, softmax assigns strictly positive attention weights to all tokens, including those with minimal relevance. As the sequence length increases, the cumulative contribution of such low-relevance tokens can introduce noise into the attention context (Martins & Astudillo, 2016).

Diffusion LLM inference repeatedly refines the same sequence across denoising steps. By focusing computation on tokens that undergo meaningful changes and suppressing contributions from non-salient positions through saliency-aware selective computation and approximation, DyLLM reduces the influence of irrelevant tokens during inference.

### 3.5. Response-only Step

Prior works (Ma et al., 2025; Wu et al., 2026a; Liu et al., 2025) reported that prompt and response tokens differ in their computational requirements. Theoretically, the relative distance decay property of Rotary Positional Embeddings (RoPE) (Su et al., 2024) imposes a locality bias on the attention mechanism. In the context of diffusion LLMs, this characteristic suggests that significant context updates tend to be spatially clustered around the latest unmasked tokens. Consequently, salient tokens are predominantly concentrated within the response tokens.

Leveraging this observation, DyLLM adopts an adaptive strategy that differentiates between prompt and response tokens, selecting salient tokens more frequently from the response than from the static prompt. In particular, during response-only steps, DyLLM does not process the full sequence at every step; instead, the input is restricted to response tokens, while the full sequence (including prompt tokens) is fed at fixed intervals (e.g., every 4 steps in this work).

In the aforementioned cache-based works, refresh steps periodically recompute all tokens in the sequence, which becomes a major throughput bottleneck. In contrast, DyLLM avoids such full token refreshes even in steps where the full sequence is provided as a model input; DyLLM restricts the expensive computation to salient tokens.

## 4. Evaluation

### 4.1. Experimental Setup

All experiments were conducted on a single NVIDIA H100 PCIe 80GB GPU. We evaluated the instruct variants of two representative open-source diffusion language models, LLaDA 8B (Nie et al., 2025) and Dream 7B (Ye et al., 2025). As baselines, we used the original implementations of the models. In addition to the original implementations, we evaluated against Fast-dLLM (Wu et al., 2026a) and dLLM-Cache (Liu et al., 2025) as strong baselines. For Fast-dLLM and dLLM-Cache, we adopted the default parameters provided by their respective implementations.

For the accuracy measurements, we applied a deterministic setting using the `lm-eval` (Gao et al., 2024) library across a diverse set of benchmarks, including mathematical reasoning (GSM8K (Cobbe et al., 2021), MATH (Hendrycks et al., 2021)), general knowledge (MMLU-Pro (Wang et al., 2024)), and code generation (MBPP (Austin et al., 2021b)). Throughput was measured as the average number of tokens generated per second.

DyLLM is implemented in PyTorch with custom CUDA kernels for sparse attention and cache operations. Native PyTorch sparse/indexing primitives introduce significant overhead at DyLLM's fine-grained token sparsity, so we implement these operations directly to realize the intended algorithmic effect. In particular, our sparse attention kernel follows a FlashAttention-like fused design (Dao et al., 2022) to expose the benefit of the proposed sparse attention pattern, while the baselines use the original FlashAttention-based attention implementations.

Unless otherwise stated, we intentionally do not apply a confidence-aware decoding (Wu et al., 2026a) scheme in our accuracy/throughput experiments to isolate the computational efficiency of each framework. The scheme operates purely at the sampling level, utilizing the model's output logits to determine tokens to decode, without modifying the underlying model execution or caching mechanisms. We evaluate compatibility with confidence-aware parallel decoding separately in Section 4.2.5.

### 4.2. Main Results

#### 4.2.1. ACCURACY PRESERVATION

We evaluate the generation quality of DyLLM against the original diffusion implementation and state-of-the-art acceleration baselines across four diverse benchmarks. Table 2 summarizes the accuracy and throughput performance on LLaDA 8B Instruct and Dream 7B Instruct compared with the baselines. To ensure deterministic accuracy evaluation, we use greedy token prediction and deterministically unmask tokens with the highest score according to each model's default setting: LLaDA uses the softmax confidence score, while Dream selects the unmasking position with the lowest entropy.

DyLLM accelerates inference by selectively recomputing

*Table 2.* Accuracy and throughput results across benchmarks with $n_u = 1$. Accuracy is shown in black, throughput (tokens per second) in blue, and speedup relative to the original implementation in orange.

| LLaDA 8B | Original | DyLLM ($\tau$=0.995) | DyLLM ($\tau$=0.99) | Fast-Prefix | Fast-Dual | dLLM-Cache |
|---|---|---|---|---|---|---|
| GSM8K | 77.79 
 11.47 (×1.00) | 78.01 
 80.15 (×6.99) | 79.08 
 87.21 (×7.60) | 77.18 
 51.05 (×4.45) | 78.24 
 75.24 (×6.56) | 77.18 
 36.77 (×3.21) |
| MBPP | 29.20 
 33.11 (×1.00) | 28.20 
 151.27 (×4.57) | 30.00 
 169.62 (×5.12) | 25.40 
 116.27 (×3.51) | 25.40 
 165.44 (×5.00) | 29.00 
 93.04 (×2.81) |
| MATH | 33.22 
 15.81 (×1.00) | 38.68 
 96.98 (×6.13) | 38.08 
 106.06 (×6.71) | 33.22 
 62.12 (×3.93) | 32.36 
 93.26 (×5.90) | 24.70 
 36.56 (×2.31) |
| MMLU-Pro | 37.31 
 11.46 (×1.00) | 37.06 
 60.16 (×5.25) | 36.42 
 66.44 (×5.80) | 36.70 
 46.45 (×4.05) | 36.18 
 64.62 (×5.64) | 37.90 
 31.56 (×2.75) |

| Dream 7B | Original | DyLLM ($\tau$=0.9975) | DyLLM ($\tau$=0.995) | Fast-Prefix | Fast-Dual | dLLM-Cache |
|---|---|---|---|---|---|---|
| GSM8K | 75.59 
 12.57 (×1.00) | 79.30 
 111.79 (×8.89) | 78.09 
 120.62 (×9.60) | 74.45 
 43.32 (×3.45) | 68.39 
 153.21 (×12.19) | 72.40 
 46.19 (×3.67) |
| MBPP | 54.60 
 19.62 (×1.00) | 56.80 
 139.62 (×7.12) | 55.40 
 141.56 (×7.22) | 53.80 
 62.96 (×3.21) | 51.60 
 205.40 (×10.47) | 52.80 
 61.06 (×3.11) |
| MATH | 37.60 
 17.64 (×1.00) | 45.12 
 130.57 (×7.40) | 43.80 
 142.34 (×8.07) | 36.48 
 59.16 (×3.35) | 36.06 
 191.56 (×10.86) | 44.98 
 48.76 (×2.76) |
| MMLU-Pro | 47.94 
 12.60 (×1.00) | 47.45 
 83.10 (×6.60) | 47.21 
 87.98 (×6.98) | 47.74 
 78.23 (×6.21) | 46.73 
 128.52 (×10.20) | 49.30 
 27.19 (×2.16) |

tokens whose attention contexts change significantly, while reusing cached activations for tokens with stable representations. Unlike fixed-schedule methods or static caching schemes that rely on rigid heuristics, DyLLM adapts the amount of computation at each layer based on token saliency, preserving critical state transitions. This allows DyLLM to maintain near-lossless accuracy across all benchmarks.

Fast-dLLM determines the recomputation region using a fixed block-based rule, which can miss critical tokens that fall outside the selected block. PrefixCache and DualCache rarely refresh the KV cache of prompt tokens, which can bias computation toward the active decoding region and miss necessary updates to previously fixed tokens including prompt tokens. In contrast, DyLLM dynamically recomputes the tokens based on their importance, maintaining higher model fidelity than existing frameworks.

While dLLM-Cache yields the highest accuracy in some cases, it suffers from the difficulty of generalization, requiring extensive tuning of hyperparameters ($K_P$, $K_R$) for each specific model and dataset (see Section D.4). Both Fast-dLLM and dLLM-Cache fix the number of computed tokens in every step before execution without prompt-awareness. This lack of prompt awareness can lead to either recomputing unnecessary stable tokens or missing tokens that become important for a particular input.

### 4.2.2. THROUGHPUT IMPROVEMENTS

DyLLM consistently delivers substantial throughput improvements across models and benchmarks in offline batched inference by dynamically concentrating computation on a small layer-dependent subset of salient tokens, avoiding redundant processing on stable tokens.

A key distinction from prior approaches is that DyLLM does not rely on periodic full sequence refresh steps. In Fast-dLLM, both PrefixCache and DualCache incur unavoidable refresh steps that recompute the entire sequence, which causes the throughput to be limited by these expensive iterations as the sequence length or parallel decoding degree increases. dLLM-Cache also has periodic refresh steps for the full sequence and for the full set of response tokens, respectively. In contrast, DyLLM maintains sparsity at every step, even when we are attending to the full sequence based on saliency.

This effect is particularly pronounced in models where FFN computation dominates runtime. Because GQA (Ainslie et al., 2023) reduces the relative cost of attention, FFN layers account for over 70% of Dream's inference time, making Dream particularly well-suited for saliency-based selective FFN execution. Consequently, DyLLM gains larger relative speedups on Dream than LLaDA, highlighting that the benefits of our method are amplified by modern attention designs such as GQA, which make attention more lightweight (see Figure 1).

Compared to dLLM-Cache, DyLLM avoids fixing the number of recomputed tokens per step. dLLM-Cache statically recomputes a predetermined fraction of tokens and relies on carefully tuned hyperparameters that vary across models and datasets. Due to this inflexibility, DyLLM is 2.16–3.67× faster than dLLM-Cache across the evaluated benchmarks. Figure 5 presents the average number of salient tokens selected by DyLLM in each layer, evaluated on 100 randomly sampled GSM8K 5-shot prompts. In DyLLM, the number of computed tokens varies flexibly across layers, resulting in substantially fewer tokens processed on average while pre-

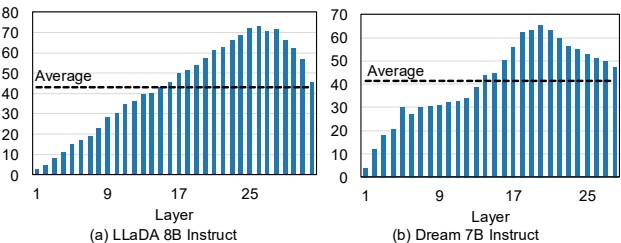

*Figure 5.* The average number of salient tokens per layer with 100 random GSM8K 5-shot prompts. $L_R$ is set to 256, as in all the other experiments. The number of salient tokens ($L_{sal}$) increases for about 3/4 of layers and starts decreasing.

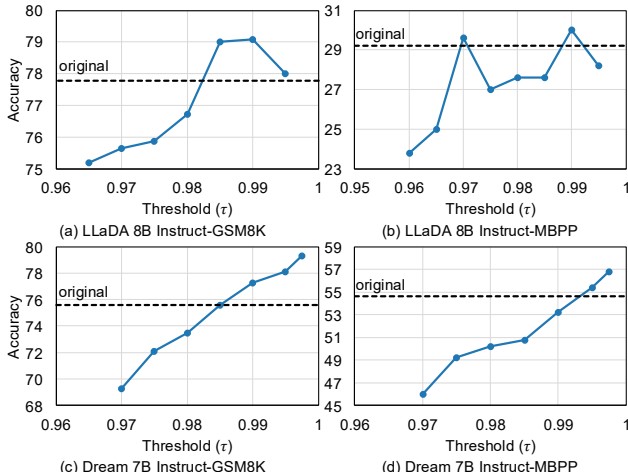

*Figure 6.* Accuracy results varying $\tau$ across GSM8K and MBPP datasets. Accuracy generally declines as the threshold is lowered.

serving accuracy. This flexibility directly results in higher throughput without sacrificing generality. Extended accuracy and throughput results on NVIDIA B200 GPUs are provided in Section D.5.

### 4.2.3. THRESHOLD $\tau$

We performed a threshold sweep to examine the trade-off between throughput and accuracy (see Figure 6). Lowering the threshold $\tau$ reduces the number of computed tokens and improves throughput; however, aggressively decreasing the threshold leads to accuracy degradation, as errors accumulate over steps without being corrected. We observed a consistent trade-off curve across models. Based on this observation, we select a threshold of 0.99 for LLaDA and 0.995 for Dream, reflecting model-specific characteristics in computational structure and sensitivity to approximation. Extended experiments for MATH and MMLU-Pro are shown in Section D.1.

### 4.2.4. SCALABILITY WITH INCREASING $n_u$

Figure 7 summarizes the accuracy-throughput trade-offs of Fast-dLLM and DyLLM as the degree of parallel decoding ($n_u$) increases. Each curve reports results for $n_u \in \{1, 2, 4\}$, marked from left to right. A critical limitation of fixed refresh schedules employed by Fast-dLLM is poor scalability. Fast-dLLM relies on periodic full refresh steps to correct approximation errors introduced by block-based caching. As sequence length or $n_u$ increases, the overhead of full refresh steps rapidly becomes dominant, significantly limiting the throughput.

This effect can be quantified by examining the average number of tokens processed per step, including refresh steps. For a representative configuration with 1024 prompt tokens and 256 response tokens, Fast-dLLM PrefixCache and Fast-dLLM DualCache compute on average 144 tokens and 32 tokens per non-refresh step, while both require processing 1280 tokens during refresh steps. With $B = 32$ and a single unmasked token per step ($n_u = 1$), the average number of computed tokens per step is $\frac{144 \times 31 + 1280}{32} = 179.5$ for PrefixCache and $\frac{32 \times 31 + 1280}{32} = 71$ for DualCache. As $n_u$ increases, the frequency of refresh steps grows correspondingly, causing the effective computation per step to

rise steeply. This problem is particularly detrimental to diffusion LLMs, whose defining advantage lies in decoding multiple tokens in parallel.

In contrast, DyLLM avoids any step that requires recomputing the entire sequence. As a result, DyLLM exhibits markedly better scalability with respect to $n_u$. This trend is consistently observed throughout Figure 7. The throughput gap between DyLLM and Fast-dLLM widens as $n_u$ grows in LLaDA. Notably, in Dream, DyLLM is slightly slower than Fast-dLLM DualCache at $n_u = 1$; however, as $n_u$ grows, DyLLM rapidly closes the gap and ultimately surpasses Fast-dLLM DualCache in throughput.

Overall, these results demonstrate that DyLLM effectively maintains the scalability characteristics that are essential to diffusion LLM inference under increasing parallel decoding without additional tuning overhead.

### 4.2.5. COMPATIBILITY WITH PARALLEL DECODING SCHEMES

Prior works have shown that adaptive parallel decoding strategies can exploit the non-autoregressive nature of diffusion LLMs more effectively than unmasking a fixed number of tokens at each step, while preserving model accuracy (Wu et al., 2026a; Jiang et al., 2026; Wu et al., 2026b; Qi et al., 2026). To examine whether DyLLM can benefit from such schemes, we further combine DyLLM with confidence-aware parallel decoding (Wu et al., 2026a) and compare it against Fast-dLLM PrefixCache and DualCache. Table 3 shows that DyLLM remains compatible with confidence-aware parallel decoding. On both LLaDA and Dream, DyLLM maintains or improves the average number of unmasked tokens per step (avg. $n_u$) while achieving higher accuracy than the Fast-dLLM-based baselines. In particular, on Dream, Fast-dLLM meaningfully degrades accuracy, whereas DyLLM preserves accuracy and even gains more from the parallel decoding scheme than the baseline.

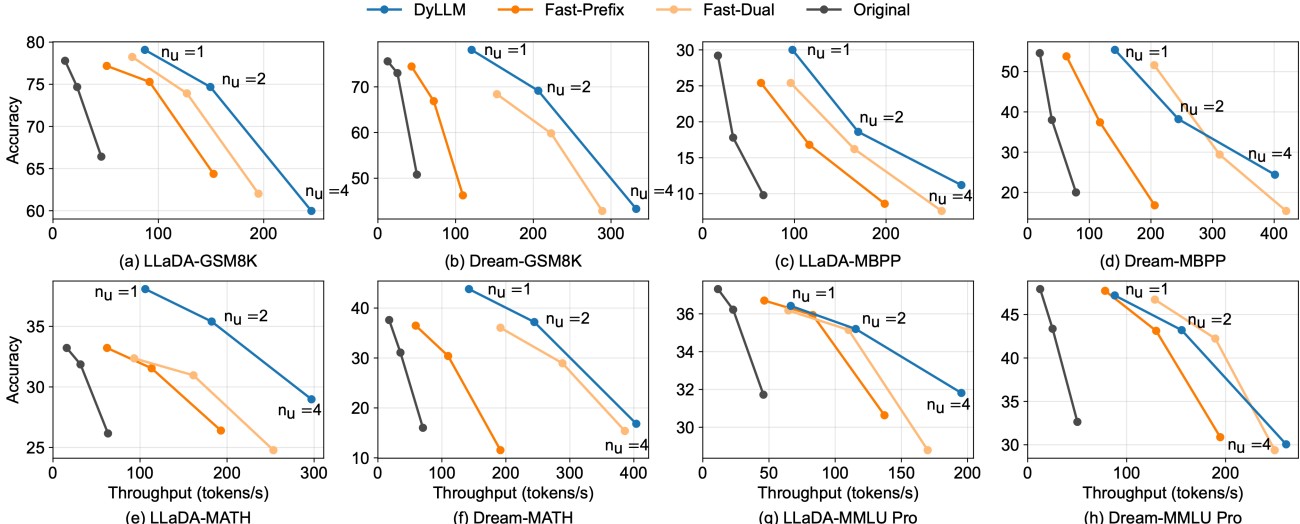

*Figure 7.* Accuracy and throughput of DyLLM, Fast-dLLM, and the original implementation across GSM8K, MBPP, MATH, and MMLU-Pro. Each curve contains three operating points corresponding to $n_u = 1, 2, 4$, where a larger $n_u$ yields higher throughput, but lower accuracy. DyLLM consistently preserves accuracy with high throughput compared to the strong baselines.

*Table 3.* Comparison of the average number of unmasked tokens per step and accuracy score on LLaDA and Dream with the GSM8K 5-shot dataset. Bold denotes the best accuracy among all methods, and underline denotes the highest Avg. $n_u$ among acceleration methods.

| Model | Method | Avg. $n_u$ | Acc. |
|---|---|---|---|
| LLaDA | Original | 3.28 | 77.86 |
| | Fast-dLLM Prefix | 3.16 | 77.94 |
| | Fast-dLLM Dual | 2.98 | 78.39 |
| | DyLLM ($\tau = 0.99$) | 3.18 | 77.03 |
| | DyLLM ($\tau = 0.995$) | 3.20 | **78.47** |
| Dream | Original | 3.82 | 75.51 |
| | Fast-dLLM Prefix | 3.77 | 73.92 |
| | Fast-dLLM Dual | 3.68 | 67.85 |
| | DyLLM ($\tau = 0.995$) | 3.89 | **78.62** |
| | DyLLM ($\tau = 0.9975$) | 3.92 | 77.10 |

These results suggest that sparse updates do not inherently conflict with parallel decoding; rather, their effectiveness depends on preserving salient token updates throughout the denoising process.

## 5. Conclusion

This paper addresses a fundamental efficiency bottleneck in diffusion LLM inference: iterative denoising requires repeatedly processing the full sequence. Our work demonstrates that the redundancy inherent in diffusion steps is not uniform but highly sparse and layer-dependent. Our framework effectively exploits this property through a layer-adaptive saliency mechanism, which dynamically isolates the small subset of tokens requiring precise updates while

approximating the remaining stable context.

Empirically, DyLLM achieves throughput improvements of up to $7.6\times$ and $9.6\times$ on LLaDA and Dream across representative benchmarks, including math, code generation, and general tasks. Crucially, unlike prior caching strategies that degrade efficiency under aggressive parallel decoding, our approach scales robustly with preserved accuracy. The computational burden of diffusion LLM inference can be significantly reduced by moving from rigid full-sequence processing to adaptive, sparsity-aware computation.

Compared with an inference framework that stores only the KV cache, DyLLM incurs additional memory overhead. Let the hidden dimension be $d$ and let $g$ be the number of query heads sharing one KV head in GQA models ($g = 1$ for MHA). Then, relative to storing only the KV cache, the memory usage increases by a factor of (2d/g + 2d) / (2d/g). However, even with state-of-the-art caching algorithms, diffusion LLMs still process tens of times more tokens per step than autoregressive LLMs. As a result, GPUs often reach peak throughput at relatively small batch sizes. In this setting, the additional memory overhead of DyLLM does not necessarily cause a comparable drop in throughput.

While DyLLM substantially reduces redundant computation in diffusion LLM inference, its performance depends on the degree of temporal sparsity exposed by the model and on the choice of the saliency threshold $\tau$. In our experiments, we identify that $\tau$ is primarily model-dependent rather than task-dependent: a single threshold calibrated once for each model generalizes well across benchmarks. Overall, DyLLM demonstrates that diffusion LLM decoding can be accelerated by exploiting temporal sparsity at the token and layer levels, without sacrificing the parallel decoding benefits of the diffusion paradigm.

## Acknowledgements

This work was partly supported by Samsung Electronics Co., Ltd (IO251211-14399-01) and IITP grant funded by MSIT (RS-2025-02304125, RS-2021-II211343, and IITP-2026-RS-2023-00256081). Jung Ho Ahn, the corresponding author, is with the Department of Intelligence and Information and the Interdisciplinary Program in Artificial Intelligence, Seoul National University.

## Impact Statement

This paper presents work whose goal is to advance the field of Machine Learning. There are many potential societal consequences of our work, none which we feel must be specifically highlighted here.

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

# A. Masked Diffusion Language Models (MDLMs)

## A.1. D3PM and MDLMs

Early attempts to apply diffusion to natural language processing primarily relied on continuous embedding diffusion (Li et al., 2022) or multinomial diffusion (Hoogeboom et al., 2021). Continuous embedding diffusion maps discrete tokens to a continuous embedding space and requires a discretization step to recover tokens, which can introduce discretization errors and semantic drift. Multinomial diffusion operates directly over discrete token states, but commonly uses uniform transition noise, where a token can be corrupted into other vocabulary entries without reflecting linguistic structure. D3PM (Austin et al., 2021a) generalized diffusion models to discrete state spaces by formulating the forward process with structured transition matrices, including uniform, nearest-neighbor, and absorbing-state transitions.

Among these choices, the absorbing-state transition is closely related to mask-based language modeling. In this formulation, the mask token is treated as an absorbing state: once a token transitions to the mask state, it remains masked for the rest of the forward process. Thus, a partially corrupted sequence consists of clean unmasked tokens and explicit mask tokens, rather than arbitrary corrupted vocabulary tokens. This provides a natural connection between discrete diffusion models and masked language modeling, where the denoising model reconstructs masked positions from bidirectional context. Modern masked diffusion language models and diffusion LLMs, such as LLaDA (Nie et al., 2025), Dream (Ye et al., 2025), and Gemini Diffusion (Google DeepMind, 2025), build on this mask-based denoising formulation.

## A.2. Diffusion Process in MDLMs

The diffusion process consists of a forward process that corrupts data and a reverse process that restores it. For MDLMs, this process is commonly instantiated with an absorbing mask state: at each step, a token either remains unchanged or transitions to the mask token according to a noise schedule $\beta_t$. In D3PM (Austin et al., 2021a), the forward process over discrete token states is defined by a transition matrix $\mathbf{Q}_t$, where $\mathbf{Q}_t(i, j)$ denotes the probability of transitioning from token $i$ to token $j$ at step $t$. Once a token reaches the mask state, it remains masked for the rest of the forward process.

The reverse process learns to invert this corruption by predicting the original tokens from a partially masked sequence. Unlike ARLMs, which generate tokens sequentially, MDLMs can predict multiple masked positions in parallel because the denoising model observes the bidirectional context of the entire partially masked sequence. For any token $i$ in the vocabulary (where $i \neq$ mask), the transition probability of step $t$ is:

$$\begin{cases} q(\mathbf{x}_t^{(i)} = [\text{mask}] | \mathbf{x}_{t-1}^{(i)}) = \beta_t \\ q(\mathbf{x}_t^{(i)} = \mathbf{x}_{t-1}^{(i)} | \mathbf{x}_{t-1}^{(i)}) = 1 - \beta_t \end{cases} \qquad t = 1 \dots T$$

where $\beta_t \in [0, 1]$ is a noise schedule that typically increases with $t$. Once a token has been changed into the mask token, it is "absorbed", meaning $q(\mathbf{x}_t^{(i)} = [\text{mask}] | \mathbf{x}_{t-1}^{(i)} = [\text{mask}]) = 1$. Because each token transitions independently, the marginal distribution $q(\mathbf{x}_t | \mathbf{x}_0)$ can be computed in closed form using the cumulative product of transition matrices $\bar{\mathbf{Q}}_t = \mathbf{Q}_1 \mathbf{Q}_2 ... \mathbf{Q}_t$. As $t \to T$, $\bar{\mathbf{Q}}_t$ converges to a state where all probability mass is on the mask token.

The reverse process $p_\theta(\mathbf{x}_{t-1} | \mathbf{x}_t)$ aims to recover the original tokens by learning the denoising distribution. The neural network, parameterized by $\theta$, is trained to approximate the posterior $p_\theta(\mathbf{x}_{t-1} | \mathbf{x}_t, \mathbf{x}_0)$.

At inference time, the learned reverse diffusion process is executed iteratively, where the model replaces masked tokens with non-mask tokens according to $p_\theta(\mathbf{x}_{t-1} | \mathbf{x}_t)$ at each diffusion step. Repeating this procedure for $T$ denoising steps, the model recovers $\mathbf{x}_0$, corresponding to a noise-free sample drawn from the learned data distribution.

# B. Diffusion LLM vs. Autoregressive LLM

## B.1. Modeling

MDLMs and autoregressive language models (ARLMs) differ in how they factorize text generation. An ARLM defines an autoregressive factorization over positions, $p_\theta(\mathbf{x}) = \prod_i p_\theta(x_i | x_{<i})$, and decoding produces the next-token distribution one position at a time. In contrast, an MDLM defines a Markov process over denoising steps, parameterizing transitions of the form $p_\theta(\mathbf{x}_{t-1} | \mathbf{x}_t)$. At each denoising step, the model outputs vocabulary logits for all positions, producing per-position categorical distributions over the sequence.

This step-wise factorization enables MDLMs to update multiple positions in a single forward pass. During inference, the model repeatedly predicts tokens for masked positions and applies an unmasking/remasking policy that keeps high-confidence predictions while refining low-confidence positions in later steps. After $T$ denoising steps, all output positions are determined.

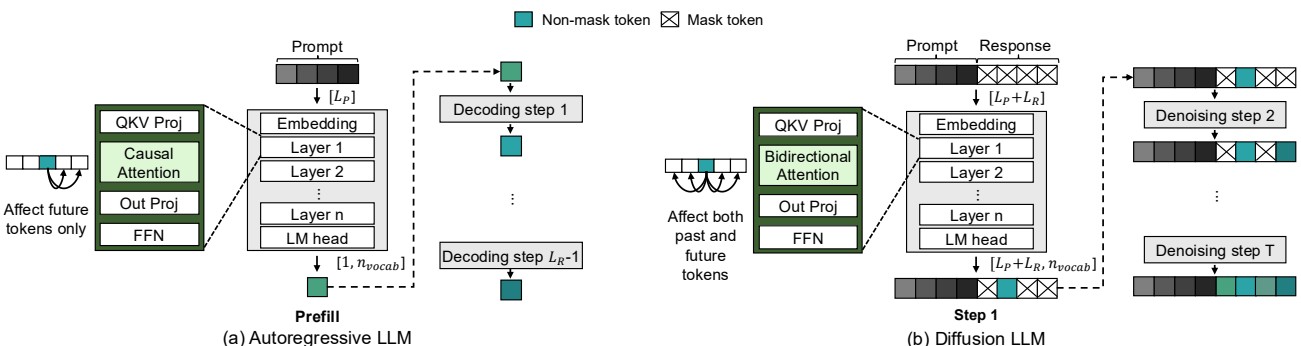

*Figure 8.* Comparison of MDLM architecture and ARLM architecture.

## B.2. Computation

Figure 8 compares the architecture of ARLMs and MDLMs. MDLMs employ a Transformer-based architecture (Vaswani et al., 2017) akin to conventional ARLMs, but exhibit two fundamental distinctions in terms of computational operations. First, MDLMs take the entire sequence—comprising both the prompt and the response—serving as input at every timestep. Similar to standard ARLMs, the MDLM architecture consists of multiple layers of self-attention and feed-forward networks (FFN) through which hidden states propagate sequentially. Consequently, the computational characteristics of a single decoding step in an MDLM are analogous to the prefill phase of an ARLM, where the entire context is processed simultaneously. The original implementation of diffusion LLM inference is highly compute-intensive, making it difficult to achieve speed comparable to that of ARLMs.

To make diffusion LLM inference faster, one option is to reduce $T$, thereby increasing the number of tokens decoded per step ($n_u = L_R/T$); however, this approach can substantially degrade response quality (Sahoo et al., 2024). With increasing model capacity, it may become feasible to substantially increase $n_u$ (Song et al., 2023), but the loss in generation quality remains unavoidable.

Second, these models utilize bidirectional attention, not causal attention. While ARLMs enforce a causal constraint, in which each token attends only to its predecessors, diffusion language models allow every token to attend to all other tokens in the sequence, both preceding and succeeding. Consequently, once a token is unmasked, all key and value representations are affected in the following step, except the first layer. This creates a divergence in activation values for the same token across different steps, meaning that applying the static KV caching mechanism used in ARLMs would lead to the accumulation of error over time.

## C. Core Algorithm of DyLLM

In this section, we provide a comprehensive breakdown of the DyLLM algorithm. The approach is designed to accelerate diffusion LLM inference by identifying *salient tokens* whose representations meaningfully changed from the previous step, and skipping redundant computations for the remainder of the sequence.

### C.1. DyLLM

Algorithm 1 presents the high-level orchestration of the generation process. Given prompt tokens $P$ of length $L_P$, DyLLM generates response tokens $R$ of length $L_R$. The response sequence $R$ is initialized with $L_R$ mask tokens, which are progressively decoded into non-mask tokens over $T_{total}$ inference steps.

DyLLM divides inference into two phases: FullStep and SparseStep. During the first $T_{full}$ steps, the model executes FullStep, which computes QKV projections, attention, output projection, and FFN for all tokens, while caching intermediate activation matrices.

After these full steps, DyLLM switches to SparseStep for the remaining steps. SparseStep reduces computation by reusing cached activations and selectively recomputing KV projections and FFN outputs only for salient tokens. For attention, SparseStep combines exact recomputation for salient tokens with approximate updates for non-salient tokens, further reducing redundant computation.

In addition, DyLLM exploits response-only steps. Since prompt tokens remain fixed throughout generation, DyLLM processes only the response sequence $R$ for response-only steps, which account for 75% of the total steps. For the remaining steps, the full sequence obtained by concatenating $P$ and $R$ is used as input.

---

**Algorithm 1** DyLLM

---

**Input:** Integer ids of prompt $P$ of length $L_P$, response length $L_R$, total number of steps $T_{total}$, number of full steps $T_{full}$, cosine similarity threshold $\tau \in \mathbb{R}$

1: Initialize $R \leftarrow [\text{mask}] * L_R$
2: Initialize $\mathbf{K_{cache}}, \mathbf{V_{cache}}, \mathbf{C_{cache}}, \mathbf{FFN\_OUT_{cache}}$
3: Initialize $\text{idx}_{\text{sal}} \leftarrow \text{None}$
4: **for** $t = 0$ **to** $T_{total} - 1$ **do**
5:    **if** $t < T_{full}$ **then**
6:       $x \leftarrow \text{Concat}(P, R)$
7:       $x \leftarrow \text{FullStep}(x, \mathbf{K_{cache}}, \mathbf{V_{cache}}, \mathbf{C_{cache}}, \mathbf{FFN\_OUT_{cache}})$
8:    **else**
9:       **if** $t \% 4 = 0$ **then**
10:          $x \leftarrow \text{Concat}(P, R)$
11:       **else**
12:          $x \leftarrow R$
13:       **end if**
14:       **if** $\text{idx}_{\text{sal}} = \text{None}$ **then**
15:          $\text{idx}_{\text{sal}} \leftarrow [\text{decoded positions}]$
16:       **end if**
17:       $x, \text{idx}_{\text{sal}} \leftarrow \text{SparseStep}(x, \mathbf{K_{cache}}, \mathbf{V_{cache}}, \mathbf{C_{cache}}, \mathbf{FFN\_OUT_{cache}}, \text{idx}_{\text{sal}}, \tau)$
18:    **end if**
19:    decoded_tokens, decoded_positions $\leftarrow \text{process\_logit}(x)$
20:    $R[\text{decoded\_positions}] \leftarrow \text{decoded\_tokens}$
21: **end for**

---

## C.2. Full Step

FullStep follows the standard transformer forward pass while augmenting it with activation caching. It takes the full sequence, including both prompt tokens $P$ and response tokens $R$, as input. Unlike KV caching in autoregressive models, FullStep additionally caches the attention context and FFN output, as shown in Algorithm 2. These caches, denoted as $\mathbf{K_{cache}}, \mathbf{V_{cache}}, \mathbf{C_{cache}}$, and $\mathbf{FFN\_OUT_{cache}}$, are reused during SparseStep to avoid redundant recomputation.

---

**Algorithm 2** FullStep

---

**Input:** Integer ids $x$ of length $L_P + L_R$, caches $\mathbf{K_{cache}}, \mathbf{V_{cache}}, \mathbf{C_{cache}}, \mathbf{FFN\_OUT_{cache}}$

1: $\mathbf{x} \leftarrow \text{embedding}(x)$
2: **for** $l = 1$ **to** $n_{layer}$ **do**
3:    $\mathbf{x} \leftarrow \text{layernorm}(\mathbf{x})$
4:    $\mathbf{Q}, \mathbf{K}, \mathbf{V} \leftarrow \text{q\_proj}(\mathbf{x}), \text{k\_proj}(\mathbf{x}), \text{v\_proj}(\mathbf{x})$    $// [L_P + L_R, d_{hidden}]$
5:    $\mathbf{C} \leftarrow \text{attention}(\mathbf{Q}, \mathbf{K}, \mathbf{V})$
6:    $\mathbf{x} \leftarrow \text{layernorm}(\text{out\_proj}(\mathbf{C}))$
7:    $\mathbf{x} \leftarrow \text{FFN}(\mathbf{x})$
8:    $\mathbf{K_{cache}}, \mathbf{V_{cache}}, \mathbf{C_{cache}}, \mathbf{FFN\_OUT_{cache}} \leftarrow \mathbf{K}, \mathbf{V}, \mathbf{C}, \mathbf{x}$
9: **end for**
10: $\text{logit} \leftarrow \text{lm\_head}(\mathbf{x})$
11: **return** logit

---

## C.3. Sparse Step

The core efficiency of DyLLM comes from SparseStep. As shown in Algorithm 3, SparseStep performs partial updates only on *salient tokens*, whose attention contexts undergo substantial drift from the previous step. At each step, SparseStep reuses activation caches from previous steps and updates only the subset of tokens identified as salient.

SparseStep first projects the hidden states of salient tokens into new key and value representations. For non-salient tokens, it reuses the cached key and value representations. The updated representations for salient tokens are then merged with the

cached representations of non-salient tokens to construct the full $\mathbf{K}$ and $\mathbf{V}$ matrices.

Using these updated KV matrices, SparseStep recomputes the attention context exactly for salient queries. For non-salient tokens, instead of recomputing the full attention output, SparseStep estimates the residual change in the attention context using approximate attention. The new attention context is then compared with the cached context via cosine similarity. Tokens whose context similarity falls below the threshold $\tau$ are marked as salient for the next layer.

Finally, SparseStep applies the FFN only to the newly identified salient tokens. For non-salient tokens, it directly reuses the cached FFN output. The updated $\mathbf{K}$, $\mathbf{V}$, attention context, and FFN output are cached for subsequent layers and steps.

---

**Algorithm 3** SparseStep

---

**Input:** Integer ids $x$ of length $L$, caches $\mathbf{K_{cache}}, \mathbf{V_{cache}}, \mathbf{C_{cache}}, \mathbf{FFN\_OUT_{cache}}$, salient token indices $\mathrm{idx_{sal}}$, cosine similarity threshold $\tau$

1: $\mathbf{x} \leftarrow \mathrm{embedding}(x)$    // $[L]$; $L$ is $L_R$ if response-only else $L_P + L_R$
2: **for** $l = 1$ **to** $n_{layer}$ **do**
3:     $\mathbf{x} \leftarrow \mathrm{layernorm}(\mathbf{x})$
4:     $\mathbf{Q} \leftarrow \mathrm{q\_proj}(\mathbf{x})$    // $[L, d_{hidden}]$
5:     $\mathbf{K}[\mathrm{idx_{sal}}] \leftarrow \mathrm{k\_proj}(\mathbf{x}[\mathrm{idx_{sal}}]), \mathbf{K}[\mathrm{idx_{sal}}^c] \leftarrow \mathbf{K_{cache}}[\mathrm{idx_{sal}}^c]$    // $[L, d_{hidden}]$
6:     $\mathbf{V}[\mathrm{idx_{sal}}] \leftarrow \mathrm{v\_proj}(\mathbf{x}[\mathrm{idx_{sal}}]), \mathbf{V}[\mathrm{idx_{sal}}^c] \leftarrow \mathbf{V_{cache}}[\mathrm{idx_{sal}}^c]$    // $[L, d_{hidden}]$
7:     $\Delta\mathbf{V} \leftarrow \mathbf{V}[\mathrm{idx_{sal}}] - \mathbf{V_{cache}}[\mathrm{idx_{sal}}]$    // $[\mathrm{len}(\mathrm{idx_{sal}}), d_{hidden}]$
8:     $\mathbf{C_{sal}} \leftarrow \mathrm{attention}(\mathbf{Q}[\mathrm{idx_{sal}}], \mathbf{K}, \mathbf{V})$    // $[\mathrm{len}(\mathrm{idx_{sal}}), d_{hidden}]$
9:     $\Delta\mathbf{C} \leftarrow \mathrm{approximate\_attention}(\mathbf{Q}, \mathbf{K}, \Delta\mathbf{V})$    // $[L, d_{hidden}]$
10:    $\mathbf{C}[\mathrm{idx_{sal}}] \leftarrow \mathbf{C_{sal}}, \mathbf{C}[\mathrm{idx_{sal}}^c] \leftarrow \mathbf{C_{cache}}[\mathrm{idx_{sal}}^c] + \Delta\mathbf{C}[\mathrm{idx_{sal}}^c]$
11:    $\mathrm{idx_{sal}} \leftarrow \mathrm{where}(\mathrm{cosine\_similarity}(\mathbf{C}, \mathbf{C_{cache}}) < \tau)$
12:    $\mathbf{x} \leftarrow \mathrm{layernorm}(\mathrm{out\_proj}(\mathbf{C}))$
13:    $\mathbf{x}[\mathrm{idx_{sal}}] \leftarrow \mathrm{FFN}(\mathbf{x}[\mathrm{idx_{sal}}]), \mathbf{x}[\mathrm{idx_{sal}}^c] \leftarrow \mathbf{FFN\_OUT_{cache}}[\mathrm{idx_{sal}}^c]$
14:    $\mathbf{K_{cache}}, \mathbf{V_{cache}}, \mathbf{C_{cache}}, \mathbf{FFN\_OUT_{cache}} \leftarrow \mathbf{K}, \mathbf{V}, \mathbf{C}, \mathbf{x}$
15: **end for**
16: $\mathrm{logit} \leftarrow \mathrm{lm\_head}(\mathbf{x})$
17: **return** $\mathrm{logit}, \mathrm{idx_{sal}}$

---

### C.4. Approximate Attention

Algorithm 4 describes how approximate attention computes the residual update of the attention context. The query matrix $\mathbf{Q}$ contains either only response tokens or the concatenation of prompt and response tokens, depending on whether the current step is response-only. In contrast, the key matrix $\mathbf{K}$ always covers the full sequence of length $L_P + L_R$.

Approximate attention first computes the attention scores by multiplying $\mathbf{Q}$ and $\mathbf{K}^\top$, followed by the softmax operation. It then extracts only the columns of the attention matrix corresponding to salient tokens. By multiplying these selected attention weights with the value residual $\Delta\mathbf{V}$ of salient tokens, approximate attention obtains the residual update $\Delta\mathbf{C}$ for the attention context. This allows DyLLM to propagate the effect of salient-token updates to all tokens without recomputing the full attention output.

---

**Algorithm 4** Approximate Attention

---

**Input:** Query $\mathbf{Q} \in \mathbb{R}^{L \times d}$, key $\mathbf{K} \in \mathbb{R}^{(L_P + L_R) \times d}$, value delta $\Delta\mathbf{V} \in \mathbb{R}^{\mathrm{len}(\mathrm{idx_{sal}}) \times d}$, salient token indices $\mathrm{idx_{sal}}$

1: $\mathbf{S} \leftarrow \mathbf{Q}\mathbf{K}^\top$    // $[L, L_P + L_R]$
2: $\mathbf{A} \leftarrow \mathrm{Softmax}(\mathbf{S})$    // $[L, L_P + L_R]$
3: $\mathbf{A}_{\mathrm{sal}} \leftarrow \mathbf{A}[:, \mathrm{idx_{sal}}]$    // $[L, \mathrm{len}(\mathrm{idx_{sal}})]$
4: $\Delta\mathbf{C} \leftarrow \mathbf{A}_{\mathrm{sal}}\Delta\mathbf{V}$    // $[L, d_{hidden}]$
5: **return** $\Delta\mathbf{C}$

---

## D. Extended Ablation Study

This section details additional results not shown in the main paper.

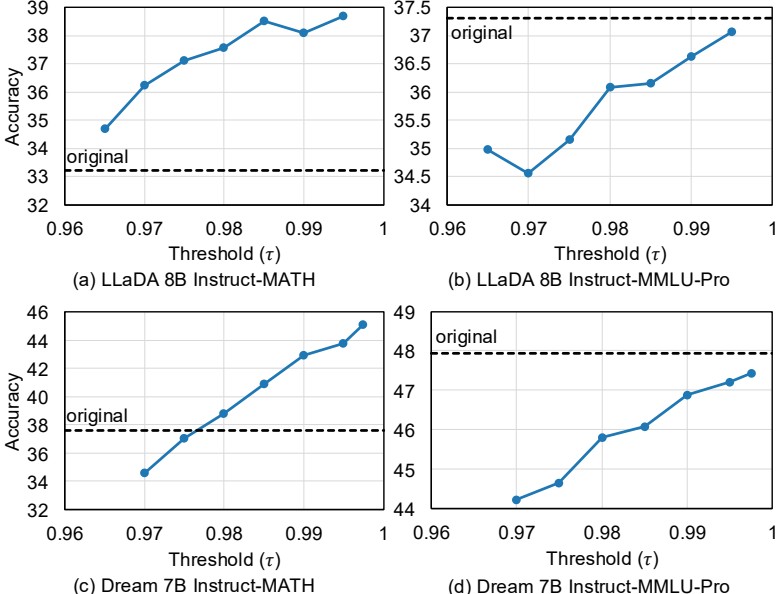

*Figure 9.* Accuracy results varying $\tau$ across MATH and MMLU-Pro benchmark.

### D.1. Impact of $\tau$ on accuracy

Extended from Figure 6, Figure 9 illustrates the accuracy on MATH and MMLU-Pro benchmarks with differing threshold values $\tau$ to examine the impact of the threshold on generation quality. We evaluated $\tau \in \{0.98, 0.985, 0.99, 0.995\}$ for LLaDA, and $\tau \in \{0.985, 0.99, 0.995, 0.9975\}$ for Dream. Since lowering $\tau$ results in selecting fewer salient tokens, thereby bypassing more computations and relying more heavily on cached results, we anticipate that a drastic decrease in $\tau$ will eventually degrade accuracy, consistent with the trend observed in Section 4.2.3.

At a finer granularity, increasing $\tau$ does not guarantee better generation quality, as it may add additional tokens irrelevant to the context. This finding is consistent with prior reports that caching frameworks (Ma et al., 2025; Wu et al., 2026a; Liu et al., 2025), including DyLLM, show better response quality than the original models in some configurations.

### D.2. Average $L_{sal}$ trend across layers and steps

Figure 10 illustrates extended results from Figure 5; the average number of salient tokens per layer across the MBPP, MATH, and MMLU-Pro benchmarks. Notably, the initial layers exhibit a markedly lower number of salient tokens.

This trend aligns with the dynamics identified in Elastic-Cache (Nguyen-Tri et al., 2026), where shallow layers are relatively more stable, exhibiting small differences between steps, whereas deeper layers continue to evolve as the effect of the updated tokens propagates to other tokens throughout the step. While Elastic-Cache computes all tokens from a certain layer onward, we flexibly select a subset of tokens at every layer, maintaining a low average number of computed tokens without compromising model accuracy.

We further observe that this tendency is consistent across inference steps. Table 4 reports the average of $L_{sal}$ at different steps for each layer. As generation progresses and fewer masked tokens remain, $L_{sal}$ generally decreases, indicating that DyLLM naturally adapts its computation to the remaining uncertainty in the sequence. Importantly, the relative layer-wise pattern of saliency remains stable across steps: shallow layers consistently require fewer salient updates, while deeper layers account for a larger fraction of the computation. This stability suggests that the proposed saliency-based selection is robust to different response lengths and numbers of denoising steps, rather than being tied to a particular generation schedule.

### D.3. Semi-AR vs. Non-Semi-AR Accuracy Comparison

Table 5 shows the effect of Semi-AR decoding on the original LLaDA and Dream implementations on the GSM8K 5-shot benchmark. The block size $B$ was set to 32. Semi-AR decoding does not alter the computations, but it yields substantial improvements in accuracy.

*Table 4.* Layer-wise average of $L_{sal}$ across steps.

| | **LLaDA** | | | | | | | | | | | | | | | |
|---|---|---|---|---|---|---|---|---|---|---|---|---|---|---|---|---|
| | GSM8K | | | | MBPP | | | | MATH | | | | MMLU-Pro | | | |
| Step | L8 | L16 | L24 | L32 | L8 | L16 | L24 | L32 | L8 | L16 | L24 | L32 | L8 | L16 | L24 | L32 |
| 0–15 | 33.4 | 50.2 | 65.4 | 23.4 | 52.2 | 63.4 | 107.4 | 54.3 | 36.3 | 45.0 | 64.4 | 28.6 | 40.1 | 56.6 | 82.8 | 34.8 |
| 64–79 | 25.5 | 48.5 | 72.8 | 43.3 | 22.3 | 49.5 | 77.5 | 46.7 | 26.6 | 46.2 | 67.6 | 40.2 | 23.3 | 58.4 | 92.7 | 56.9 |
| 128–143 | 21.6 | 43.0 | 59.4 | 40.9 | 22.8 | 46.3 | 68.9 | 46.7 | 23.8 | 44.1 | 63.2 | 42.1 | 20.8 | 45.9 | 71.8 | 44.4 |
| 192–207 | 17.8 | 36.3 | 51.3 | 41.8 | 19.2 | 41.6 | 66.3 | 50.2 | 20.3 | 38.4 | 59.7 | 48.5 | 17.3 | 39.2 | 68.0 | 49.4 |

| | **Dream** | | | | | | | | | | | | | | | |
|---|---|---|---|---|---|---|---|---|---|---|---|---|---|---|---|---|
| | GSM8K | | | | MBPP | | | | MATH | | | | MMLU-Pro | | | |
| Step | L7 | L14 | L21 | L28 | L7 | L14 | L21 | L28 | L7 | L14 | L21 | L28 | L7 | L14 | L21 | L28 |
| 0–15 | 28.6 | 40.0 | 71.6 | 35.2 | 40.1 | 65.4 | 89.3 | 66.0 | 30.2 | 54.8 | 83.6 | 48.5 | 45.2 | 72.4 | 99.9 | 56.6 |
| 64–79 | 42.9 | 52.5 | 77.4 | 54.3 | 35.9 | 45.5 | 64.5 | 52.9 | 33.2 | 49.7 | 78.9 | 50.3 | 47.1 | 56.9 | 83.8 | 61.2 |
| 128–143 | 36.7 | 46.0 | 64.5 | 48.8 | 22.2 | 28.3 | 48.7 | 31.2 | 41.4 | 52.3 | 73.3 | 50.5 | 48.5 | 56.4 | 84.3 | 58.0 |
| 192–207 | 25.6 | 34.3 | 54.4 | 39.4 | 22.4 | 23.9 | 34.4 | 25.6 | 35.3 | 42.1 | 61.6 | 45.6 | 42.1 | 50.7 | 79.1 | 55.4 |

*Table 5.* Impact of Semi-AR decoding on accuracy score

| | **LLaDA** | | **Dream** | |
|---|---|---|---|---|
| | **Semi-AR** | **Non-Semi-AR** | **Semi-AR** | **Non-Semi-AR** |
| $n_u = 1$ | 77.79 | 58.07 | 75.59 | 42.30 |
| $n_u = 2$ | 74.68 | 54.36 | 67.12 | 40.41 |
| $n_u = 4$ | 66.41 | 50.19 | 48.45 | 39.12 |

### D.4. dLLM-Cache Configurations

Table 6 details the configuration settings to reproduce the dLLM-Cache baseline. The hyperparameters $K_P$ and $K_R$ denote the prompt refresh interval and the response refresh interval, respectively. The values of $K_P$ and $K_R$ differ by an order of magnitude, resulting in frequent full computation over the response tokens, which might be helpful for long prompts. We adopted the identical values reported in the original paper, as the baseline's static refresh policy requires dataset-specific tuning for optimal performance.

*Table 6.* Interval steps for LLaDA Instruct and Dream Instruct of dLLM-Cache

| Model | Interval | GSM8K | MBPP | MATH | MMLU-Pro |
|---|---|---|---|---|---|
| LLaDA 8B Instruct | $K_P$ | 50 | 100 | 50 | 51 |
| | $K_R$ | 7 | 7 | 1 | 3 |
| Dream 7B Instruct | $K_P$ | 25 | 10 | 50 | 5 |
| | $K_R$ | 2 | 8 | 1 | 1 |

### D.5. Accuracy and Throughput Results on NVIDIA B200

Table 7 reports accuracy and throughput results for DyLLM and Fast-dLLM on NVIDIA B200 using a batch size of 16. The results confirm that the main conclusion of our H100 evaluation also holds on the newer Blackwell-generation GPU.

## E. Formal Proofs of Propositions

In this section, we provide the detailed mathematical derivations for the propositions presented in Section 3.2. These proofs establish the theoretical foundation for using temporal cosine similarity as a proxy for salient token selection in MDLMs.

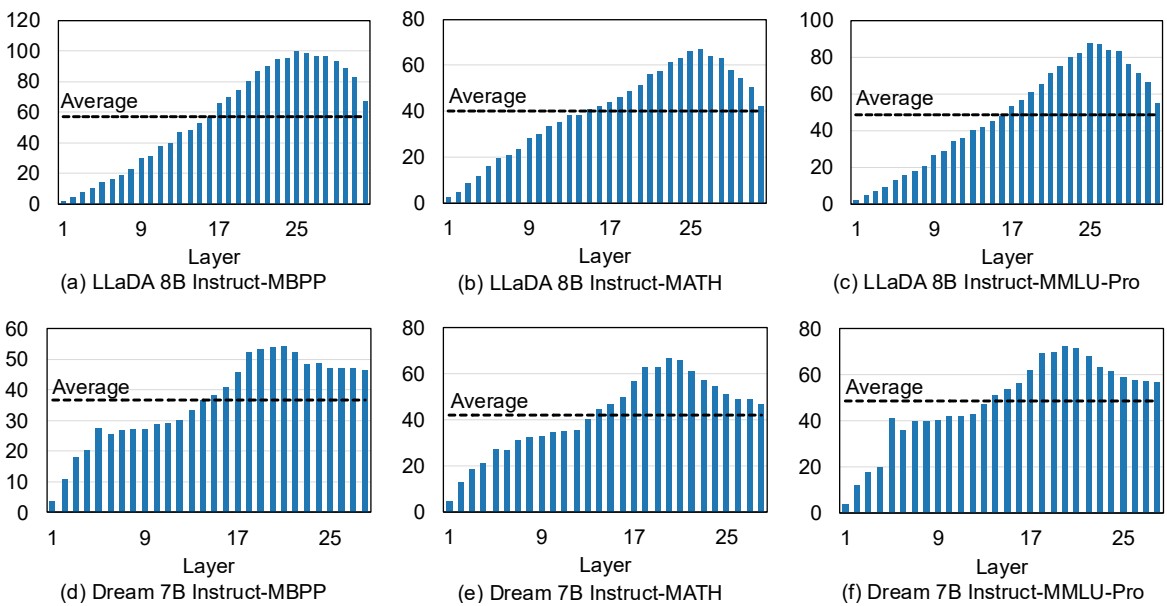

*Figure 10.* The average of $L_{sal}$ per layer across MBPP, MATH, and MMLU-Pro. The same layer-wise trend observed on GSM8K also appears across these benchmarks.

*Table 7.* Accuracy and throughput results across benchmarks with $n_u = 1$ on NVIDIA B200. Accuracy is shown in black, throughput (tokens per second) in blue, and speedup relative to the original implementation in orange.

| LLaDA 8B | Original | DyLLM ($\tau$=0.995) | DyLLM ($\tau$=0.99) | Fast-Prefix | Fast-Dual |
|---|---|---|---|---|---|
| GSM8K | 76.80
31.27 ($\times$1.00) | 77.10
184.81 ($\times$5.91) | 78.09
196.24 ($\times$6.28) | 77.18
94.85 ($\times$3.03) | 78.09
126.71 ($\times$4.05) |
| MBPP | 29.40
43.11 ($\times$1.00) | 30.00
210.92 ($\times$4.89) | 29.00
227.29 ($\times$5.27) | 25.20
112.71 ($\times$2.61) | 26.00
143.69 ($\times$3.33) |
| MATH | 33.46
44.41 ($\times$1.00) | 38.98
226.55 ($\times$5.10) | 37.92
244.28 ($\times$5.50) | 33.06
124.38 ($\times$2.80) | 32.70
160.41 ($\times$3.61) |
| MMLU-Pro | 37.06
26.93 ($\times$1.00) | 36.57
147.32 ($\times$5.47) | 35.36
157.58 ($\times$5.85) | 36.87
86.95 ($\times$3.23) | 36.58
118.53 ($\times$4.40) |
| **Dream 7B** | **Original** | **DyLLM ($\tau$=0.9975)** | **DyLLM ($\tau$=0.995)** | **Fast-Prefix** | **Fast-Dual** |
| GSM8K | 76.12
29.62 ($\times$1.00) | 79.68
237.76 ($\times$8.03) | 78.39
248.79 ($\times$8.40) | 73.92
169.02 ($\times$5.71) | 69.14
255.09 ($\times$8.61) |
| MBPP | 54.00
47.11 ($\times$1.00) | 54.60
282.49 ($\times$6.00) | 56.40
300.62 ($\times$6.38) | 54.00
184.10 ($\times$3.91) | 51.80
271.88 ($\times$5.77) |
| MATH | 38.20
45.50 ($\times$1.00) | 44.82
278.23 ($\times$6.11) | 44.20
293.13 ($\times$6.49) | 37.52
215.9 ($\times$4.75) | 36.30
342.58 ($\times$7.53) |
| MMLU-Pro | 47.92
26.05 ($\times$1.00) | 47.59
192.53 ($\times$7.39) | 47.63
204.23 ($\times$7.84) | 47.18
165.29 ($\times$6.35) | 46.69
248.59 ($\times$9.54) |

## E.1. Proof of Proposition 3.1

**Proposition 1 (Scale Invariance under Linear Projection).** *Let $C \in \mathbb{R}^d$ be the attention context, $W_o \in \mathbb{R}^{d \times d}$ be the output projection matrix, and $\alpha \in \mathbb{R}^+$ be a positive scaling factor. The composite operation of linear projection followed by RMSNorm satisfies: $RMSNorm((\alpha C)W_o) = RMSNorm(CW_o)$.*

*Proof.* By the linearity of the output projection $W_o$, the projected representation scales linearly with the input:

$$(\alpha C)W_o = \alpha(CW_o). \tag{4}$$

Let $Y = CW_o$ denote the projected vector. Recall the definition of the RMSNorm operation:

$$\text{RMSNorm}(Y) = \frac{Y}{\text{RMS}(Y)} \odot \mathbf{g}, \quad \text{where} \quad \text{RMS}(Y) = \sqrt{\frac{1}{d} \sum_{i=1}^{d} y_i^2}. \tag{5}$$

Applying this to the scaled vector $\alpha Y$:

$$\text{RMS}(\alpha Y) = \sqrt{\frac{1}{d} \sum (\alpha y_i)^2} = \sqrt{\alpha^2 \cdot \frac{1}{d} \sum y_i^2} = |\alpha|\text{RMS}(Y). \tag{6}$$

Since $\alpha \in \mathbb{R}^+$, we have $|\alpha| = \alpha$. Substituting these into the RMSNorm formula:

$$\text{RMSNorm}(\alpha Y) = \frac{\alpha Y}{\alpha \text{RMS}(Y)} \odot \mathbf{g} = \frac{Y}{\text{RMS}(Y)} \odot \mathbf{g} = \text{RMSNorm}(Y). \tag{7}$$

Thus, $\text{RMSNorm}((\alpha C)W_o) = \text{RMSNorm}(CW_o)$, completing the proof. $\square$

### E.2. Proof of Proposition 3.2

**Proposition 2 (Error Bound via Directional Alignment).** *The $L_2$ error between normalized outputs, $\delta = \|\text{RMSNorm}(Y_t) - \text{RMSNorm}(Y_{t-1})\|_2$, is bounded by the temporal cosine similarity $s_{t,l}$ of the inputs as:*

$$\delta \leq \kappa(W_o)\sqrt{2(1 - s_{t,l})}. \tag{8}$$

*Proof.* Let $\hat{C}_t$ and $\hat{C}_{t-1}$ be the unit-normalized input vectors (i.e., $\hat{C} = C/\|C\|$). The temporal cosine similarity is given by the dot product $s_{t,l} = \hat{C}_t^\top \hat{C}_{t-1}$. The Euclidean distance between these unit inputs is strictly determined by $s_{t,l}$:

$$\|\hat{C}_t - \hat{C}_{t-1}\|_2 = \sqrt{2(1 - s_{t,l})}. \tag{9}$$

Consider the function $f(x) = \frac{W_o x}{\|W_o x\|_2}$ which maps a unit vector $x$ to the normalized output space. The approximation error $\delta$ corresponds to the change in this map's output. Standard results in numerical linear algebra establish that the local Lipschitz constant of the map $x \mapsto Wx/\|Wx\|$ on the unit sphere is bounded by the condition number $\kappa(W_o)$. Specifically, for any unit vectors $u, v$:

$$\left\| \frac{W_o u}{\|W_o u\|} - \frac{W_o v}{\|W_o v\|} \right\|_2 \leq \kappa(W_o)\|u - v\|_2. \tag{10}$$

Applying this inequality to our inputs $\hat{C}_t, \hat{C}_{t-1}$ and outputs $\hat{Y}_t, \hat{Y}_{t-1}$ (where $\hat{Y}$ is the normalized output):

$$\delta = \|\hat{Y}_t - \hat{Y}_{t-1}\|_2 \leq \kappa(W_o)\|\hat{C}_t - \hat{C}_{t-1}\|_2. \tag{11}$$

Substituting the expression for the input distance:

$$\delta \leq \kappa(W_o)\sqrt{2(1 - s_{t,l})}. \tag{12}$$

This confirms that the upper bound holds strictly without relying on small-angle approximations. $\square$

## F. Other Related Works

### F.1. Confidence Aware Decoding

Confidence-aware decoding (Wu et al., 2026a) employs a dynamic decoding strategy that is independent of the predefined number of denoising steps $T$. Specifically, it unmasks any positions whose confidence score exceeds a user-specified threshold during the generation process. This approach represents a distinct optimization scope compared to architectural or caching strategies. It operates strictly at the sampling level, using the model's output logits to determine which tokens to decode. Because this optimization is model-agnostic, it can be seamlessly integrated not only with Fast-dLLM but also with our proposed method and any other aforementioned frameworks.

## F.2. Sparse Attention

To address the quadratic complexity of attention operations in AR LLMs, various sparse attention mechanisms have been proposed. These methods typically leverage the observation that attention scores are highly localized, with tokens primarily attending to their neighbors (Child et al., 2019; Beltagy et al., 2020). Beyond static patterns, recent works have focused on runtime sparsity, analyzing dynamic attention patterns to prune less critical computations during inference. For instance, H2O (Zhang et al., 2023) and Deja Vu (Liu et al., 2023) discard tokens with low accumulated attention scores (KV cache eviction) or predict contextual sparsity to skip blocks, while MInference (Jiang et al., 2024) identifies specific sparse patterns to accelerate the prefill stage.

While these methods primarily explore sparsity in autoregressive models, similar ideas have also been applied to diffusion LLMs. During inference of diffusion LLMs, only a small subset of tokens changes at each step, leading to substantial redundancy in attention score activations across adjacent steps. SparseD (Wang et al., 2026) exploits this property by computing full attention in early steps and, after a pre-defined step, constructing a binary attention mask by selecting high-scoring attention blocks. This mask is then reused in subsequent steps, allowing the model to perform efficient sparse operations for the remainder of the denoising steps.

