# OpenReview forum: "DyLLM: Efficient Diffusion LLM Inference via Saliency-based Token Selection and Partial Attention"
_ICML.cc/2026/Conference — ICML 2026 regular_

### Official Review · Reviewer_gWNu · 2026-03-07

**Soundness:** 2
**Presentation:** 3
**Significance:** 2
**Originality:** 2
**Overall Recommendation:** 4
**Confidence:** 4

**Summary:**

the authors propose DyLLM -- an inference time technique to speed up diffusion language models, building on the self-tested hypothesis that token representations in diffusion language models remain largely static across timesteps, and only a small number of tokens differ significantly. Using this idea, they only selectively update the token representations in attention computation, and skip FFN processing for the non-salient tokens, as they call them. Their empiricial results depict speedups on standard benchmarks, using LLADA and Dream as their representative base models, while being quality neutral, or with slight improvements/drops in quality.

**Compliance With Llm Reviewing Policy:**

Affirmed.

**Final Justification:**

The rebuttal largely answered my empirical questions, but there are still some missing pieces and arguments that need work to justify certain results. I have increased my score by 1 point and hope that the authors will work on the missing pieces as well.

**Key Questions For Authors:**

- In proposition 3.2, it is said that as long as the temporal variations between the attn context vectors at consecutive timesteps is "small" -- what does small mean here -- not well formalized
- As mentioned in the above section as well, how is aggressive pruning for earlier layers and more lenient for later ones achieved? Is that just an artifact of the threshold based selection of salient tokens? The fig 7 does have an initial increasing trend but it also drops down towards the end, suggesting intermediate layers having most salient tokens.
- Why does accuracy improve for certain tasks using DyLLM? Can the softmax norm logic be ablated? If that is indeed the cause, why does accuracy drop for certain other kind of tasks?
- While fig 7 shows the number of salient tokens per layer, how does this trend vary across timesteps (after the initial T_full phase)
- It would be nice if the authors can also share some of their forward looking research directions and limitations of their current approach in the conclusion section.

**Limitations:**

No, it is not well discussed, and as per my points above, i highly recommend the authors to do so.

**Strengths And Weaknesses:**

1. soundness:
- The method is well motivated using an empirical analysis that shows that most token representations remain largely similar based on the attention score similarity heurisitc.
- The choice of attention score similarity for judging whether token representations update is not very well discussed in my opinion. As per my thinking, the tokens that attended to other tokens in a certain fashion in previous iterations will continue to do so largely, while only small changes will occur to this behaviour. Whether this immediately suggests that token representations within a certain threshold bound need not be updated is not very clear.
- Around Line 150 in the right section (just above Sec 3), the authors note that other methods for speeding up diffusion language modeling rely on "fixed schedules or global thresholds" -- however DyLLM itself has a threshold which needs tuning. It is not very clear what the threshold should be. The authors use the accuracy-speed tradeoff as a heurisitc to figure out the thresholds. However, this might also vary for different tasks, and not just different models, which is not discussed in the paper. On a related note, LInes 210 (right side) mention that the pruning is more aggressive in earlier layers and less aggresive later, but in the empirical section, only a single global threshold is discussed.
- In sec 3.4, the authors mention modest improvements to quality, with softmax normalization being the issue intuitively in the naive approach. However, there are no relevant ablations/theory to back this up in the paper.
- Sec 3.5 discusses locality bias, and mentions that for Diffusion LMs, it means that context updates tend to be spatially clustered around the latest unmasked tokens -- again, there are no ablations or theoretical grounding in the paper to back this up. The authors also keep the prompt tokens in the domain for choosing salient tokens for 25% of the sparse steps, which is only discussed in the appendix and not well ablated.
- IN Figure 5, there are accuracy drops with increasing threshold $\tau$, the reason behind which is neither clear nor discussed.

2. presentation:
- The presentation is largely nice, and the authors present a coherent narrative, mention relevant previous literature, and compare against a good number of relevant baselines. There are certain statements (mentioned in my "soundness" section) which need clarity.

3. significance:
- The method provides a somewhat more general version of the inference time approaches that have come up for accelerating diffusion language models, not redoing global computations, and varying the number of relevant tokens per layer.
- The existance of a global (or potentially layerwise and timestepwise threshold) to indentify salient tokens restrict the neatness of the approach to some extent. While still helpful, it might require a careful selection of tau for each task and model variant, which might lead to limited generalization.
- As per the results, the speedups are significant, and in a range comparable to other speedup methods. However there are still accuracy drops at certain tasks, and the method doesnt globally beat other inference methods for an approach that is supposed to be more general. Importantly, the figure 6 results suggest that as the number of parallel decoded tokens are increased, for a lot of datasets, the accuracy does sharply drop as compared to the baseline, while for a few it remains neutral.

4. originality:
- While the idea of pruning and conditional computation is not new, the inference time selection, and the application to DLMS in the form that it is done is still original.

---

> ### Author Rebuttal · Authors · 2026-03-31
>
> **Justification of temporal similarity-based judgement**
>
> To clarify, we compare the cosine similarity of the attention context vectors (score·V), not the attention scores themselves.
> Proposition 3.2 bounds the approximation error  δ in the normalized attention output as: δ ≤ κ(W_o) √(2(1 − s_{t,l})). When cosine similarity exceeds τ, the FFN input is close to that of the previous step, so reusing the cached FFN output introduces only bounded, threshold-controlled error. Furthermore, since the FFN computation is token-wise independent, skipping one token's update does not affect others within the same layer. Plus, Proposition 3.2 does not require "small timestep" condition to keep it valid. We will omit it in revision.
>
> **Why a single τ is still adaptive**
> Prior methods such as Fast-dLLM and dLLM-Cache use fixed recomputation rules (e.g., a fixed block or every K steps) that are set before execution and do not depend on the model’s runtime state. In contrast, DyLLM uses a threshold τ to decide recomputation dynamically at each layer and step.
>
> The table below reports temporal cosine-similarity percentiles across layers: pK means K% of tokens have cosine similarity ≤ pK. At τ = 0.99, for example, LLaDA Layer 8 has p5 = 0.998, so fewer than 5% of tokens are selected. Figure 7 directly reflects this: the same single global τ naturally allocates more computation to more dynamic layers without layer-specific tuning.
>
> |LLaDA  | p5 | p10 | p50 |
> |---|---:|---:|---:|
> | Layer 8  | 0.998 | 0.999 | 1.000 |
> | Layer 16 | 0.991 | 0.997 | 1.000 |
> | Layer 24 | 0.953 | 0.980 | 1.000 |
> | Layer 32 | 0.967 | 0.987 | 1.000 |
>
> | Dream | p5 | p10 | p50 |
> |---|---:|---:|---:|
> | Layer 7  | 0.997 | 0.999 | 1.000 |
> | Layer 13 | 0.978 | 0.994 | 1.000 |
> | Layer 20 | 0.961 | 0.987 | 1.000 |
> | Layer 27 | 0.977 | 0.990 | 0.999 |
>
> **τ tuning per task?**
> We tested on math (GSM8K, MATH), coding (MBPP), and general knowledge (MMLU-pro). The same τ per model works well across all tasks. The main variation is across models (0.99 for LLaDA, 0.995 for Dream), not tasks. In practice, τ can be set once per model using a small validation set, similar to how quantization calibration is done.
>
> **Spatial locality of salient tokens**
> We measured span (distance between first and last salient token) and compactness (num salient tokens / span) on GSM8K  with response-only steps disabled:
>
> **LLaDA**
>
> | Layer | Avg Span | Avg Compactness |
> |---|---:|---:|
> | 8  | 36.0 | 0.56 |
> | 16 | 25.4 | 0.64 |
> | 24 | 36.1 | 0.52 |
> | 32 | 31.1 | 0.60 |
>
> **Dream**
>
> | Layer | Avg Span | Avg Compactness |
> |---|---:|---:|
> | 7  | 18.7 | 0.81 |
> | 14 | 24.5 | 0.64 |
> | 21 | 32.3 | 0.56 |
> | 28 | 32.6 | 0.62 |
>
> Given context lengths of ~1K tokens, spans of 18–36 confirm that salient tokens are tightly clustered. This is consistent with prior works (Fast-dLLM, dLLM-Cache, Elastic-Cache), though pushing this too aggressively hurts accuracy (e.g., Dream GSM8K: 75.59 → 68.39 with DualCache).
>
> **Accuracy improvement**
>
> While DyLLM is an approximation, it acts as a thresholded gate that suppresses low-magnitude temporal updates. If $(\Delta S_{t,l})V_{t-1,l}$ bypassing it can improve the denoising signal-to-noise ratio, explaining why selective computation can occasionally outperform dense baselines (e.g., the GSM8K gains).
>
> This aligns with a broader phenomenon where filtering low-information components—seen in dropout, pruning, and recent mechanisms like SpargeAttention (Zhang et al., 2025)—preserves or even enhances performance. Similar accuracy gains reported in other diffusion LLM works (e.g., Fast-dLLM, $D^2$Cache, SparseD) further suggest this is a general property of selective inference rather than a DyLLM-specific artifact.
>
> To verify quality preservation, we measured unmasking confidence on GSM8K. As shown below, confidence remains stable across τ, supporting that DyLLM does not degrade—and can sometimes improve—the effective decision signal.
>
> |  τ | LLaDA Unmask Conf. | Dream Unmask Conf. |
> | :--- | :---: | :---: |
> | 1.0 (Original) | 0.988 | 0.950 |
>  | 0.995 | 0.987 | 0.946 |
> | 0.99 | 0.987 | 0.946 |
> | 0.985 | 0.987 | 0.941 |
> | 0.98 | 0.985 | 0.941 |
>
> **Avg number of salient token trends across steps per layer**
> Two patterns emerge. First, layers with more salient tokens in Figure 7 remain the most active throughout. Second, salient-token counts may rise temporarily in early steps, but later stabilize and decrease as more tokens become fixed. Importantly, the count does not grow unboundedly over time.
>
> **LLaDA**
>
> | Steps \ Layer | 7 | 15 | 23 | 31 |
> |---|---:|---:|---:|---:|
> | 0–15 | 33.4 | 50.2 | 65.4 | 23.4 |
> | 64–79 | 25.5 | 48.5 | 72.8 | 43.3 |
> | 128–143 | 21.6 | 43.0 | 59.4 | 40.9 |
> | 192–207 | 17.8 | 36.3 | 51.3 | 41.8 |
>
> **Dream**
>
> | Steps \ Layer | 6 | 13 | 20 | 27 |
> |---|---:|---:|---:|---:|
> | 0–15 | 28.7 | 40.0 | 71.6 | 35.2 |
> | 64–79 | 42.9 | 52.5 | 77.4 | 54.3 |
> | 128–143 | 36.7 | 46.0 | 64.5 | 48.8 |
> | 192–207 | 25.6 | 34.3 | 54.4 | 39.4 |

---

> > ### Author Rebuttal · Reviewer_gWNu · 2026-04-04
> >
> > Thank you for the detailed rebuttal. Certain questions of mine have been well answered (attn context vectors, prop 3.2 error, $tau$ stability, span/compactness metrics, timestep wise salient token).
> >
> > Things still not addressed:
> > - softmax normalization ablation: unmasking confidence numbers and  analogies to dropout are suggestive only. Related, accuracy drop for certain tasks, the symmetry question, are still unanswered.
> > - fig 5 accuracy drops with increasing tau for certain tasks still go unaddressed
> > - prompt token inclusion question is not answered
> > - limitations - not answered
> >
> > I would recommend the authors try to update the draft with these answers, meanwhile, given the current empirical justifications, i am happy to increase my score by 1 point.

---

> > > ### Author Response · Authors · 2026-04-08
> > >
> > > Thank you for the thoughtful follow-up and for carefully engaging with our rebuttal. We appreciate your constructive feedback and the score increase, and we respond to the remaining points below.
> > >
> > > **softmax ablation**
> > >
> > > “Salient” in Table 1 denotes the variant that applies salient token selection only to FFN execution, without approximate attention applying full attention to DyLLM. “Salient+Approx” denotes the full DyLLM method, where both salient token selection and approximate attention are enabled. The results suggest that assigning non-zero attention scores to all tokens at every step through softmax normalization is not a strict requirement for perserving accuracy.
> > >
> > > **accuracy drop with higher $\tau$ and for certain task**
> > >
> > > We added further experiments to better understand the observed accuracy fluctuations. For the softmax distribution $p$ at each position and step, $E_{umk}$ is the average entropy over positions unmasked at that step and HC-Flip is the fraction of (step, position) pairs where the predicted token changes between consecutive steps despite high confidence in the previous step. We measured $E_{umk}$ and HC-Flip differing $\tau$ and the results in the table below shows that DyLLM preserves entropy at a level similar to the original model, while consistently reducing HC-Flip across both LLaDA and Dream.
> > > This suggests that DyLLM does not increase uncertainty, but instead yields a more stable decoding trajectory with fewer unnecessary high confidence reversals. The effect is especially pronounced on the LLaDA MBPP experiment, where DyLLM achieves both lower entropy and lower HC-Flip than the original model, which may contribute to the accuracy improvement. Consistent with the paper, we claim that DyLLM effectively preserves the model’s accuracy, not necessarily increasing accuracy for all the tasks.
> > >
> > > | Model | Task | $\tau$ | E_umk | HC Flip |
> > > |---|---|---:|---:|---:|
> > > | LLaDA | GSM8K | original | 0.068 | 0.16% |
> > > |  |  | 0.995 | 0.077 | 0.11% |
> > > |  |  | 0.99 | 0.075 | 0.08% |
> > > |  |  | 0.985 | 0.074 | 0.07% |
> > > |  |  | 0.98 | 0.076 | 0.06% |
> > > |  | MBPP | original | 0.054 | 0.11% |
> > > |  |  | 0.995 | 0.043 | 0.07% |
> > > |  |  | 0.99 | 0.048 | 0.05% |
> > > |  |  | 0.985 | 0.060 | 0.05% |
> > > |  |  | 0.98 | 0.066 | 0.05% |
> > > | Dream | GSM8K | original | 0.156 | 0.09% |
> > > |  |  | 0.9975 | 0.163 | 0.07% |
> > > |  |  | 0.995 | 0.167 | 0.05% |
> > > |  |  | 0.99 | 0.171 | 0.03% |
> > > |  |  | 0.985 | 0.175 | 0.02% |
> > > |  | MBPP | original | 0.070 | 0.02% |
> > > |  |  | 0.9975 | 0.072 | 0.02% |
> > > |  |  | 0.995 | 0.073 | 0.02% |
> > > |  |  | 0.99 | 0.071 | 0.01% |
> > > |  |  | 0.985 | 0.097 | 0.01% |
> > >
> > >
> > > **Response-only step (prompt token inclusion)**
> > > DyLLM periodically includes prompt tokens in the refresh steps at a configurable period. As shown in the span/compactness analysis, salient tokens are spatially gathered near unmasked tokens, which supports the use of response-only steps. We ablate the full refresh step period (N) on LLaDA ($\tau=0.99$) with the GSM8K dataset:
> > > | Period (N) | 1 | 2 | 4 | 8 | 16 |
> > > | Accuracy | 75.36 | 77.86 | 79.08 | 76.88 | 77.10 |
> > > Accuracy remains stable across settings. The optimal N can vary depending on factors such as input/output length and prompt characteristics, so we choose a relatively conservative value. In particular, more frequent prompt inclusion reduces throughput by expanding the token selection scope, while the marginal throughput benefit of increasing N becomes smaller at larger periods, following the same diminishing returns trend moving from ¼ to ⅛ compared with ½ to 1/4 . Empirically, $N=4$ provides a good balance between accuracy and efficiency, and we therefore use it across all experiments without per-task tuning.
> > >
> > > **Limitation**
> > > In addition to the KV cache, DyLLM maintains a context cache and an FFN output cache. The context cache is needed to identify salient tokens based on temporal similarity, and the FFN output cache is needed to support layer-wise adaptive token selection. Compared with an inference framework that only stores the KV cache, DyLLM incurs additional memory overhead. Let the hidden dimension be d and the number of query heads sharing one KV head be g in GQA models (g=1 for MHA models). Then, relative to storing only the KV cache, the memory overhead increases by a factor of
> > > \frac{2d/g + 2d}{2d/g}.
> > > This overhead may be less restrictive in diffusion LLMs than it might initially seem. Even with state-of-the-art caching algorithms, diffusion LLMs still process tens of times more tokens per step than autoregressive LLMs. As a result, GPUs often reach peak throughput at relatively small batch sizes. In this setting, the additional memory overhead of DyLLM does not necessarily cause a comparable drop in throughput.

---

### Official Review · Reviewer_5DJj · 2026-03-11

**Soundness:** 3
**Presentation:** 2
**Significance:** 3
**Originality:** 4
**Overall Recommendation:** 4
**Confidence:** 4

**Summary:**

This paper introduces DyLLM, a novel training-free inference framework for diffusion LLMs, aiming at addressing the challenge of the expensive computation brought by bidirectional attention. Specifically, the authors discover that for the majority of tokens, the attention context remains highly consistent between consecutive iterations, and introduce a dynamic selection policy that identifies salient tokens
at each layer. As a result, they design a specialized sparse attention and caching mechanism by selectively computing only these salient tokens. By extensive evaluations on different benchmarks and comparison with other baselines, DyLLM achieves comparable efficiency with strong baselines, while preserving better generation quality.

**Compliance With Llm Reviewing Policy:**

Affirmed.

**Key Questions For Authors:**

1. In Figure 2, what does the "Counts (normalized)" mean? In the 2D slice at each step, is the Counts (normalized)-cosine similarity curve something like a CDF that indicates the proportion of tokens with cosine similarity smaller than a value? Could the authors further describe Figure 2 in detail?

1. What do "Salient" and "Salient + Approx." mean in Table 1?

1. How does DyLLM refresh the KV cache? According to the last paragraph in Section 3.5, DyLLM performs computation only on salient tokens at refresh steps. To my understanding, this is the same as the denoising step. If so, why does DyLLM still need stand-alone refresh steps?

1. What is the detailed configuration of the token sampling and unmasking? Could the authors provide more details on it, including
    - How is each token sampled from the distribution? By temperature sampling or greedy sampling? And based on which score (e.g., entropy or softmax confidence)?
    - How are the unmasked positions selected? By selecting the positions with higher scores?

1. Could the authors provide a more detailed description of Figure 3? For example, what are the white blocks? The queries of non-salient tokens? And what are the diagonal striped blocks and dashed blocks?

1. Besides, in Figure 3, are all the keys used to compute the column sparse attention? Or only keys of the salient tokens, i.e. $K_{sal}$ (if following the denotation of row sparse attention)?

1. What is the definition of $\Delta V$?

1. To my understanding, DyLLM still requires computing queries for all tokens, but can skip some computation in the attention and FFN layers. Is that correct? And in addition to the KV cache, does DyLLM also have to maintain an attention output cache ($C_{old}$) and a FFN output cache? If so, what is the additional memory overhead?

1. Although the authors intentionally exclude the confidence-aware parallel decoding[1], I am curious how DyLLM will impact the parallel decoding capability of diffusion LLMs. For example, when using the same confidence threshold, will more or fewer tokens be accepted per step under the confidence-aware parallel decoding?

### Reference
1. Wu, Chengyue, et al. "Fast-dllm: Training-free acceleration of diffusion llm by enabling kv cache and parallel decoding." arXiv preprint arXiv:2505.22618 (2025).

**Limitations:**

1. The memory overhead compared with other baselines is not discussed.

1. As mentioned in Questions, the compatibility with parallel decoding algorithms can be further studied in the future.

**Strengths And Weaknesses:**

### Strengths
1. The whole structure of the paper is clear to me.
1. The proposed method is intuitive and reasonable. It introduces an adaptive sparse attention and caching mechanism for diffusion LLMs, beyond fixed schedules or global thresholds that do not account for the layer-wise dynamics, and provides valuable insights to the community.
1. The proposed method is evaluated by extensive results on different models and benchmarks, and compared with different baselines. From the empirical evaluations, it achieves comparable efficiency with strong baselines like Fast-dllm Dual Cache, while preserving better generation quality.

### Weaknesses
1. Some figures and tables do not provide informative captions. Please see the Questions below.
1. Some experimental configurations are not clear enough. Please see the Questions below.
1. Overall, the presentation of this paper is not clear enough and makes it hard to fully understand this paper, although the proposed method is solid to me.

---

> ### Author Rebuttal · Authors · 2026-03-31
>
> **Figure/Table, Configuration clarification** \
> Thank you for the careful reading and helpful feedback. We agree that several fixes in figures, tables, and experimental descriptions will improve the readability of our work, and we will revise the captions, main text, table cell labels, and detailed experimental configurations accordingly.
>
> For Figure 2, each 2D slice at a given denoising step shows the distribution of temporal cosine similarity values $s_{t,l}^{(i)}$ across tokens, where each value compares the attention context vector of token $i$ at step $t$ with that of the same token at step $t-1$. We also agree that the y-axis label should be changed to “Fraction of Tokens” to make the figure more explicit.
>
> For Table 1, we also need to clarify the terminology. DyLLM consists of two components:
>
> (1) salient-token-only FFN execution based on temporal similarity, and
>
>  (2) approximate attention.
>
> “Salient” in Table 1 refers to applying only salient-token selection for FFN execution, without approximate attention, whereas “Salient + Approx.” refers to the full DyLLM method with both components enabled. In the revision, we will rename these entries more explicitly.
>
> For Figure 3 and the related questions, the white blocks denote non-salient tokens, and the diagonal striped blocks denote the $\Delta$ terms. For column-sparse attention, all keys are still needed to compute the score columns corresponding to salient tokens, while only the value vectors of salient tokens are needed to compute the sparse attention output.
>
> $\Delta V$ denotes the difference from the previous step, i.e., $V_{t,l} - V_{t-1,l}$, analogous to the definition of $\Delta C$ in Section 3.3.
>
> Regarding token sampling and unmasking, we used greedy sampling and selected positions with higher scores for unmasking to keep the accuracy evaluation deterministic. The unmasking criterion follows each model’s default setting: LLaDA uses confidence score, while Dream uses entropy.
>
> **Refresh step** \
> The last paragraph of Section 3.5 may have caused confusion. To clarify, DyLLM itself does not require a separate refresh step that fully recomputes all tokens. The full-token refresh described there was introduced for Fast-dLLM and dLLM-Cache in the later part, rather than DyLLM. Your understanding is correct: DyLLM still computes queries for all input tokens in a step. The savings do not come from skipping query generation, but from the saliency-aware algorithm.
>
> **Confidence-aware Parallel Decoding** \
> Thank you for this question. To further examine this point, we additionally implemented confidence-aware parallel decoding on top of DyLLM and compared it against the original implementation and Fast-dLLM baselines under the same decoding scheme (confidence threshold 0.9, following Fast-dLLM). The results show that DyLLM preserves the benefit of parallel decoding. This confirms that DyLLM is compatible with confidence-aware decoding for further acceleration.
>
> | LLaDA | original | PrefixCache | DualCache | DyLLM 0.99 | DyLLM 0.995 |
> |---|---:|---:|---:|---:|---:|
> | Avg. unmasked token per step | 3.28 | 3.16 | 2.98 | 3.18 | 3.20 |
> | Accuracy | 77.86 | 77.94 | 78.39 | 77.03 | 78.47 |
>
> | Dream | original | PrefixCache | DualCache | DyLLM 0.995 | DyLLM 0.9975 |
> |---|---:|---:|---:|---:|---:|
> | Avg. unmasked token per step | 3.82 | 3.77 | 3.68 | 3.89 | 3.92 |
> | Accuracy | 75.51 | 73.92 | 67.85 | 78.62 | 77.10 |
>
> As noted in Section 4.1, we view parallel decoding as a sampling-level strategy, whereas Fast-dLLM, dLLM-Cache, and DyLLM target model-side computation reduction. Thus, we considered it important to isolate these two effects in the main paper. We will include these additional compatibility results in the final version.
>
> **Limitations** \
> Thank you for this important point. We agree that memory overhead should be discussed more explicitly, and will do so in Section 6.
>
> In addition to the KV cache, DyLLM maintains a context cache and an FFN output cache. The context cache is needed to identify salient tokens based on temporal similarity, and the FFN output cache is needed to support layer-wise adaptive token selection. Thus, compared with an inference framework that stores only the KV cache, DyLLM incurs additional memory overhead. Let the hidden dimension be $d$ and let $g$ be the number of query heads sharing one KV head in GQA models ($g=1$ for MHA). Then, relative to storing only the KV cache, the memory usage increases by a factor of $\frac{2d/g + 2d}{2d/g}$.
>
> This overhead may be less restrictive in diffusion LLMs than it might initially seem. Even with state-of-the-art caching algorithms, diffusion LLMs still process tens of times more tokens per step than autoregressive LLMs. As a result, GPUs often reach peak throughput at relatively small batch sizes. In this setting, the additional memory overhead of DyLLM does not necessarily cause a comparable drop in throughput.

---

> > ### Author Rebuttal · Reviewer_5DJj · 2026-04-01
> >
> > Thank the authors for the response. I think a refined presentation and a discussion about memory overhead in the future version will further improve this paper. Overall, I think this is a technically solid paper. I will maintain my recommendation for weak accept.

---

> > > ### Author Response · Authors · 2026-04-08
> > >
> > > Thank you for the positive feedback and for acknowledging that your concerns have been fully addressed. We also appreciate your suggestions on refining the presentation and further discussing memory overhead, and we will incorporate them in the final version.

---

### Official Review · Reviewer_hZPt · 2026-03-11

**Soundness:** 2
**Presentation:** 3
**Significance:** 3
**Originality:** 2
**Overall Recommendation:** 4
**Confidence:** 3

**Summary:**

The paper proposes DyLLM, a training-free inference framework designed to accelerate Masked Diffusion Language Models (MDLMs). MDLMs suffer from high computational costs because they repeatedly process the entire sequence at every denoising step, effectively performing a "repeated prefill". DyLLM identifies "salient tokens" by measuring the cosine similarity of attention contexts between steps. It then recomputes FFN and attention operations only for these tokens while reusing cached activations for stable ones. The authors report up to 9.6x higher throughput on models like LLaDA and Dream while maintaining accuracy. While the idea is interesting, it has two major issues: 1. it looks a bit incremental wrt to other papers and 2. it is not clear how the speedup was obtained in practical implementation.

**Compliance With Llm Reviewing Policy:**

Affirmed.

**Final Justification:**

It was not clear to me the authors implemented a CUDA kernel since they never mentioned it in the paper, only in the rebuttal. Without that, it was hard to justify the efficiency improvements. Now that they have clarified, I have increased my score accordingly.

**Key Questions For Authors:**

- Did you implement custom kernels to realize the wall-clock speedups ? If not, how do you actually get faster execution given that pytorch is not natively optimized to leverage sparsity ?
- Was the "Original" baseline implemented with any standard optimizations, or is the speedup compared against a completely unoptimized "repeated prefill" execution?
- Since DyLLM does not use periodic full refreshes , how does the framework prevent the accumulation of approximation errors over a high number of diffusion steps?
- Can you elaborate on "We intentionally did not apply a confidence-aware decoding (Wu et al., 2025) scheme in our experiments to isolate the computational efficiency of each framework." ?

**Limitations:**

yes

**Strengths And Weaknesses:**

### **Strengths**
- Good presentation
- (originality, soundness) The paper identifies "layer-wise temporal sparsity" in Masked Diffusion Language Models (MDLMs), revealing that early layers remain highly stable during iterative denoising while deeper layers evolve.
- (Soundness) The authors provide mathematical error bounds (Propositions 3.1 and 3.2) to justify using temporal cosine similarity as a proxy for token saliency.

### **Weaknesses**
- (major, soundness) While the paper claims per-step runtime reductions, it lacks detail on the underlying software implementation. Standard PyTorch operations for sparse indexing and "Column Sparse Attention" typically incur overhead on GPUs that can negate theoretical gains.I am wondering where the speedup actually comes from, or if it is just theoretical.
- In several cases, the approximation framework reportedly outperforms the dense baseline. For instance, LLaDA on GSM8K rises from 77.79 to 78.85 when using salient-only FFNs.
- The saliency threshold ($\tau$) requires model-specific tuning, such as 0.99 for LLaDA versus 0.995 for Dream, limiting the "training-free" ease of use. It looks like the method is quite sensible to hyperparam tuning

---

> ### Author Rebuttal · Authors · 2026-03-31
>
> **(major) Software implementation**
>
> Thank you for these important questions. We first clarify that the reported speedups are measured end-to-end wall-clock speedups on a single NVIDIA H100 PCIe 80GB GPU (Section 4.1), not theoretical projections. Figure 1 shows the runtime breakdown observed in practice. We address your major concerns below:
>
> **Speedup source**: The main gain comes from reducing the dominant FFN/projection computation by processing only salient tokens, along with response-only steps that reduce the number of tokens processed in a step (Fig 1). The FFN and FC layers are not specially accelerated; they are implemented with standard PyTorch (nn.Linear), identical to the baselines. In the original implementation, FFN in a step used to take 417.38 us (49.78%), 594.32 us (77.40%)  reduced to 18.91 us and 17.28 us respectively in LLaDA and Dream (batch size 16).
>
> **Sparsity in PyTorch**: We agree that naïve PyTorch sparse indexing would introduce substantial overhead and would likely fail to realize algorithmic gains. Therefore, we implemented the sparse attention path with a custom CUDA kernel using a FlashAttention-style blocking design. We also added lightweight custom handling for cache access needed by the sparse path. To further clarify the system contribution and implementation details, we have also provided an anonymized code repository in the submission.
>
> **Baseline fairness**: The “Original” baseline uses the model’s officially released codebase with FlashAttention enabled and the standard repeated-prefill execution, i.e., the same baseline used by Fast-dLLM and dLLM-Cache.
>
> **Regarding novelty**: DyLLM is, to the best of our knowledge, the first diffusion LLM inference framework that uses layer-adaptive saliency to jointly reduce both attention and FFN computation. Prior approaches rely on fixed token selection or caching rules; DyLLM instead combines saliency-based adaptive computation with saliency-aware approximate attention.
>
> We will revise the implementation section to clarify these design choices explicitly.
>
> **Accuracy improvement**
>
> While DyLLM is an approximation, it effectively acts as a threshold gate that suppresses low-magnitude temporal updates. If $\Delta S_{t,l}$ serves as a low-SNR component, bypassing it can improve the denoising signal-to-noise ratio, explaining why selective computation can occasionally outperform dense baselines (e.g., the GSM8K gains).
>
> This aligns with a broader phenomenon where filtering low-information components—seen in dropout, pruning, and recent mechanisms like SpargeAttention (Zhang et al., 2025)—preserves or even enhances performance. Similar accuracy gains reported in other diffusion LLM works (e.g., Fast-dLLM, Elastic-Cache, $D^2$Cache, SparseD) further suggest this is a general property of selective inference rather than a DyLLM-specific artifact.
>
> To verify quality preservation, we measured unmasking confidence on GSM8K. As shown below, confidence remains stable across τ, supporting that DyLLM does not degrade—and can sometimes improve—the effective decision signal.
>
> | $\tau$ | LLaDA Unmask Conf. | Dream Unmask Conf. |
> | :--- | :---: | :---: |
> | 1.0 (Original) | 0.988 | 0.950 |
> | 0.995 | 0.987 | 0.946 |
> | 0.99 | 0.987 | 0.946 |
> | 0.985 | 0.987 | 0.941 |
> | 0.98 | 0.985 | 0.941 |
>
> **Confidence-aware Parallel Decoding**
>
> We implemented confidence-aware parallel decoding on top of DyLLM and compared it against the original and Fast-dLLM baselines under the same decoding scheme (confidence threshold set to 0.9 following Fast-dLLM paper). The results show that DyLLM preserves the benefit of the parallel decoding. This confirms that our method can be combined with confidence-aware decoding for further acceleration.
>
> | LLaDA | original | PrefixCache | DualCache | DyLLM 0.99 | DyLLM 0.995 |
> |---|---:|---:|---:|---:|---:|
> | Avg. unmasked token per step | 3.28 | 3.16 | 2.98 | 3.18 | 3.20 |
> | Accuracy | 77.86 | 77.94 | 78.39 | 77.03 | 78.47 |
>
> | Dream | original | PrefixCache | DualCache | DyLLM 0.995 | DyLLM 0.9975 |
> |---|---:|---:|---:|---:|---:|
> | Avg. unmasked token per step | 3.82 | 3.77 | 3.68 | 3.89 | 3.92 |
> | Accuracy | 75.51 | 73.92 | 67.85 | 78.62 | 77.10 |
>
> We view parallel decoding as a sampling-level strategy, whereas Fast-dLLM, dLLM-Cache, and DyLLM target model-side computation reduction (Sec 4.1); we therefore isolated these two effects in the paper and will include the additional compatibility results in the final version.
>
> **$\tau$ tuning**
> We tested on math (GSM8K, MATH), coding (MBPP), and general knowledge (MMLU-pro). The same τ per model works well across all tasks. The main variation is across models, not tasks. In practice, τ can be set once per model using a small validation set, similar to how quantization calibration is done.
> We agree, however, that this calibration requirement is a practical limitation relative to a fully plug-and-play method, and we will make this trade-off explicit in the revised paper.

---

> > ### Author Rebuttal · Reviewer_hZPt · 2026-04-03
> >
> > Thank you for your answers, I appreciate your replies and will increase my score to 4.

---

> > > ### Author Response · Authors · 2026-04-08
> > >
> > > Thank you for the thoughtful follow-up and for carefully considering our response. We sincerely appreciate your updated assessment and score. We will make sure to incorporate the clarified implementation details, additional experiments, and the discussion of practical trade-offs more clearly in the revised version.

---

### Official Review · Reviewer_xSYk · 2026-03-12

**Soundness:** 3
**Presentation:** 3
**Significance:** 3
**Originality:** 3
**Overall Recommendation:** 5
**Confidence:** 4

**Summary:**

This paper studies the decoding process of Masked Diffusion Language Models and observes that, between two adjacent decoding steps, only a small number of salient tokens exhibit significant changes in their representations. Based on this observation, authors propose a sparse decoding algorithm, DyLLM, built on salient token selection. In DyLLM, attention computation is performed only for the selected salient tokens, while the representations of other tokens are approximately updated using the changes in the value vectors of the salient tokens. In addition, the FFN layers are computed only for these salient tokens, which substantially reduces the overall computation cost. Experiments show that DyLLM achieves significant acceleration on LLaDA 8B and Dream 7B across multiple tasks while largely preserving model performance.

**Compliance With Llm Reviewing Policy:**

Affirmed.

**Final Justification:**

The authors have added sufficient experiments, addressing my main concern regarding long-context settings, and have responded to my other questions point by point. Therefore, I have adjusted my score.

**Key Questions For Authors:**

1. Since Figure 7 shows that salient tokens are distributed unevenly across layers, would it make sense to use different $\tau$ for different layers?

2. In Table 2, it would improve readability if some of the most important values were highlighted in bold.

3. What do authors think is the reason that, in Figure 5, the performance of LLaDA 8B at $\tau=0.99$ is better than at $\tau=1$, whereas for Dream 7B the trend appears monotonic?

**Limitations:**

yes

**Strengths And Weaknesses:**

**Strengths**:

1. *The paper is logically well structured.* It first introduces cosine similarity between representation vectors as the criterion for salient token selection, then analyzes the distribution of salient tokens to demonstrate their sparsity. Building on this observation, the paper proposes the sparse decoding algorithm DyLLM. Authors also carefully decompose the temporal change in the attention context and introduce approximate attention to compensate for the loss caused by not fully recomputing attention for non-salient tokens.
2. *The empirical results are strong.* On the provided benchmarks, the method achieves both better speedup and better performance than previous acceleration approaches, and it appears to be close to the Pareto frontier.
3. *The analysis is relatively thorough.* The paper includes both theoretical and empirical error analyses, providing a per-step error bound as well as the CDF of the approximate attention error, which to some extent strengthens the credibility of the method. In addition, the ablation study on the threshold $\tau$ for salient token selection is fairly sufficient.

**Weaknesses:**

1. DyLLM does not process the full sequence at every step, but instead only does so once every fixed number of steps. Although this does not seem to hurt performance on the benchmarks considered in the paper, these benchmarks mostly involve relatively short inputs or cases where the answer depends only weakly on input context. I am concerned that for other types of tasks, such as long-context tasks where the response depends heavily on knowledge distributed throughout the input, this design choice may lead to substantial performance degradation.
2. Although the paper provides a single-step error analysis, the accumulation of error remains unclear for more complex tasks with longer reasoning chains that require more denoising steps. While it may be difficult to establish a full theoretical guarantee, empirical evidence on the scale of this accumulated error is needed.

---

> ### Author Rebuttal · Authors · 2026-03-31
>
> **Long Context**
>
> Thank you for raising this important concern. We agree that robustness of (i) infrequent access to the prompt during response-only steps, (ii) approximate attention, and (iii) salient-token-only FFN execution should also be shown in long-context settings. To examine this more directly, we added experiments on both long-input and long-output settings.
>
> For long-input evaluation, we used RULER-4K tasks, and for long-output evaluation, we additionally evaluated BBH CoT zeroshot subtasks with the maximum output length set to 1K tokens. We selected subtasks that require long reasoning and whose response depends more on the input prompt than on few-shot prompts. The results show that DyLLM still largely preserves model accuracy in both settings compared to the original and periodic refresh algorithm.
>
> These additional results suggest that the approximation error does not grow uncontrollably even in settings with stronger prompt dependence or longer reasoning chains. We will include these long-input and long-output experiments in the final version to better support the robustness of DyLLM beyond the original benchmark suite.
>
> | task | original | DyLLM | PrefixCache | DualCache |
> |---|---:|---:|---:|---:|
> | ruler_qa_squad | 88.30 | 87.07 | 83.57 | 82.43 |
> | ruler_qa_hotpot | 83.00 | 83.60 | 80.20 | 80.40 |
> | ruler_cwe | 68.80 | 69.08 | 67.40 | 70.76 |
> | ruler_vt | 97.40 | 96.24 | 92.84 | 92.60 |
> | niah_multiquery | 100.00 | 100.00 | 99.85 | 99.80 |
>
> | task | original | DyLLM | PrefixCache | DualCache |
> |---|---:|---:|---:|---:|
> | bbh_cot_zeroshot_multistep_arithmetic_two | 65.20 | 66.40 | 67.20 | 68.80 |
> | bbh_cot_zeroshot_reasoning_about_colored_objects | 81.20 | 81.20 | 78.40 | 75.20 |
> | bbh_cot_zeroshot_tracking_shuffled_objects_seven_objects | 14.40 | 15.20 | 14.80 | 16.40 |
>
> **Trend difference between LLaDA and Dream**
>
> We would first like to clarify a point about Figure 5: the case of $\tau=1$ is not explicitly plotted as a curve; instead, the black dashed line indicates the accuracy of the original dense implementation, which corresponds to $\tau=1$, so DyLLM also showed slight improvement on Dream, not only LLaDA.
>
> Nonetheless, we agree that the trends of LLaDA and Dream differ in the figure. When we further increased the $\tau$ to 0.999 in Dream with the GSM8K dataset, the accuracy score dropped from 79.23 ($\tau=0.9975$) to 78.92 matching the trend of LLaDA. This might be task-specific behavior, but another plausible reason is their difference in attention architecture. As Dream uses GQA, where a key/value head is shared across a group of query heads, it may exhibit a different tolerance to approximation than models with standard MHA.
>
> **Different $\tau$ for different layers**
>
> Thank you for this insightful suggestion. We agree that layer-wise thresholding is a promising direction. As shown in Figure 2, the temporal similarity distribution varies across layers, indicating that the degree of changes across tokens is layer-dependent. Figure 7 can be viewed as a direct consequence of this observation: under a fixed threshold $\tau$, DyLLM naturally selects different numbers of salient tokens across layers, adapting to the layer-wise variation in token dynamics.
>
> Layer-specific thresholds can be set by calibrating on a small validation set, to further improve compute efficiency by allocating computation more adaptively across layers. We did not explore this in the current work, however, because it would introduce additional hyperparameter tuning overhead. We therefore adopt a single global threshold in this paper, while viewing layer-wise threshold calibration as a promising direction for future work.
>
> For Table 2, we will make sure to put important values in bold in the revision.

---

> > ### Author Rebuttal · Reviewer_xSYk · 2026-04-06
> >
> > The authors have added sufficient experiments, addressing my main concern regarding long-context settings, and have responded to my other questions point by point. Therefore, I have adjusted my score.

---

> > > ### Author Response · Authors · 2026-04-08
> > >
> > > Thank you for the careful follow-up. We appreciate that you found the additional long-input and long-output experiments helpful in addressing your main concern. We will incorporate these results, along with the clarifications on layer-wise thresholding and the LLaDA/Dream trend difference, more clearly in the revised version.

---

### Decision · Program_Chairs · 2026-04-30

**Decision:**

Accept (regular)

**Comment:**

This paper proposes DyLLM, a training-free framework that accelerates Masked Diffusion Language Model inference by selectively updating only "salient tokens" based on temporal cosine similarity. During the review process, the authors successfully addressed concerns regarding hardware efficiency by demonstrating a 9.6x wall-clock speedup using custom CUDA kernels and provided extensive new results confirming robustness in long-context settings. Reviewers also explored the adaptive nature of the threshold-based selection, the spatial locality of token updates, and the trade-offs of additional memory overhead, ultimately concluding that the method effectively stabilizes decoding trajectories while significantly reducing computation. With scores of 5, 4, 4, and 4, there is a clear consensus that the paper is technically sound and provides a practical, high-impact contribution to efficient LLM inference.